# Steering Large Language Models for Machine Translation with Finetuning and In-Context Learning

**Duarte M. Alves**[1,4] **Nuno M. Guerreiro**[1,2,4,5] **João Alves**[2] **José Pombal**[2]
**Ricardo Rei**[2,3,4] **José G. C. de Souza**[2] **Pierre Colombo**[5,6] **André F. T. Martins**[1,2,4]

[1]Instituto de Telecomunicações, Lisbon, Portugal [2]Unbabel, Lisbon, Portugal,
[3]INESC-ID, Lisbon, Portugal [4]Instituto Superior Técnico, University of Lisbon, Portugal
[5]MICS, CentraleSupélec, Université Paris-Saclay, France [6]Equall, Paris, France
duartemalves@tecnico.ulisboa.pt

## Abstract

Large language models (LLMs) are a promising avenue for machine translation (MT). However, current LLM-based MT systems are brittle: their effectiveness highly depends on the choice of few-shot examples and they often require extra post-processing due to overgeneration. Alternatives such as finetuning on translation instructions are computationally expensive and may weaken in-context learning capabilities, due to overspecialization. In this paper, we provide a closer look at this problem. We start by showing that adapter-based finetuning with LoRA matches the performance of traditional finetuning while reducing the number of training parameters by a factor of 50. This method also outperforms few-shot prompting and eliminates the need for post-processing or in-context examples. However, we show that finetuning generally *degrades* few-shot performance, hindering adaptation capabilities. Finally, to obtain the best of both worlds, we propose a simple approach that incorporates few-shot examples *during* finetuning. Experiments on 10 language pairs show that our proposed approach recovers the original few-shot capabilities while keeping the added benefits of finetuning.[1]

## 1 Introduction

Large language models (LLMs) have shown remarkable performance on a wide range of NLP tasks by leveraging in-context learning (Brown et al., 2020). In particular, when provided with few-shot examples, these models have demonstrated impressive capabilities for performing machine translation (MT) without requiring explicit supervision on parallel data (Garcia et al., 2023). However, this approach exhibits several drawbacks: performance is highly dependent on the quality of examples (Vilar et al., 2022), outputs are plagued by overgeneration (Bawden and Yvon, 2023), and inference costs

---

[1]Code avaliable at https://github.com/deep-spin/translation_llm.

are greatly increased by processing all input pairs. When parallel data is available, LLMs can alternatively be finetuned on translation instructions (Li et al., 2023). This method generally outperforms few-shot prompting and eliminates the need for in-context examples. However, it remains unclear whether finetuned models can benefit from the desirable properties of in-context learning, such as on-the-fly domain adaptation (Agrawal et al., 2022). Additionally, traditional finetuning (Devlin et al., 2019; Radford et al., 2018) incurs a high computational overhead due to the cost of updating all the model weights.

In this paper, we provide a closer examination of the impact of finetuning and few-shot prompting for adapting LLMs to perform translation. Our experiments encompass 10 language pairs on general and specific domains, comprising over 100,000 generated translations (§2). Our main findings are:

- We show that finetuning with adapters (Houlsby et al., 2019; Hu et al., 2022) is a very effective method to steer LLMs for translation (§3.1). This method matches the performance of traditional finetuning at a fraction of the computational cost, by training 50 times fewer parameters. It also achieves better translation quality than in-context learning and eliminates the need for post-processing the generated outputs and selecting in-context examples.

- We show that finetuning large language models degrades their few-shot performance, limiting their adaptation capabilities (§3.2). In particular, we show that finetuned LLMs perform poorly on domain adaptation scenarios when provided in-context examples.

- To address this issue, we propose a simple approach that introduces few-shot examples *during* finetuning (§4). Our results show that we can recover few-shot capabilities while retaining the benefits of finetuning.

## 2 Experimental Setup

In our experiments, we use LLaMA 7B and 13B (Touvron et al., 2023) as backbone language models and finetune them with the standard cross entropy loss.

We train our models on general domain OPUS (Tiedemann, 2012) data from the Europarl, Globalvoices, Paracrawl, Tilde, Ubuntu, and Wikipedia domains. We consider the languages Dutch (nl), French (fr), German (de), Portuguese (pt) and Russian (ru), both from and into English (en).[2] To ensure the quality of the training records, we first apply Bicleaner (Ramírez-Sánchez et al., 2020) using a threshold of 0.85 and then filter the remaining pairs, ensuring both language directions have a COMETKiwi (Rei et al., 2022b) score above 0.8. Finally, we sample 250K records for each language pair. During training, we uniformly sample from the data to ensure each language pair is seen a similar number of times. We perform validation on the Flores-200 development set for the language pairs in the training data.

For in-domain evaluation, we consider the Flores-200 (NLLB Team et al., 2022) test dataset on all the translation directions included during training, as well as the WMT22 test sets[3] for the language pairs considered in our training data. Regarding data for specialized domains, we consider the Medical and Law domains from Aharoni and Goldberg (2020), the TICO dataset (Anastasopoulos et al., 2020) and WMT Chat (Farinha et al., 2022). We evaluate our models on zero and five shot settings, uniformly sampling for each test sentence five independent few-shot samples from the respective development set.

Our main evaluation metric is COMET (Rei et al., 2020, 2022a)[4]. We also report results with BLEU (Papineni et al., 2002), chrF (Popović, 2015) and COMETKiwi (Rei et al., 2022b) in Appendix G.

We refer the reader to Appendix A for full details on hyperparameters and instruction formats used in the following experiments.

[2]We also consider Chinese (zh). However, as it is not supported by LLaMA, we examine it in Appendix B.

[3]https://www.statmt.org/wmt22/translation-task.html

[4]We use the latest COMET model wmt22-comet-da from version 2.0.1.

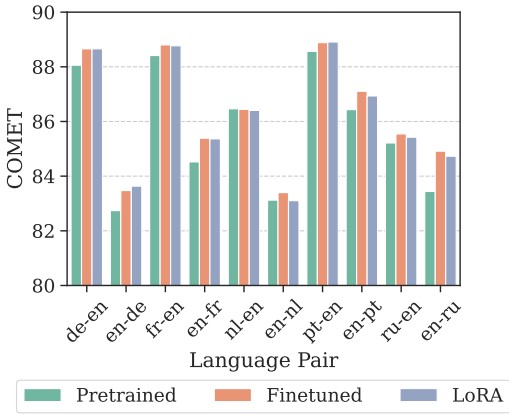

Figure 1: COMET scores on the Flores-200 test set by LLaMA 7B pretrained (few-shot) and LLaMA 7B trained with full finetuning and LoRA (zero-shot).

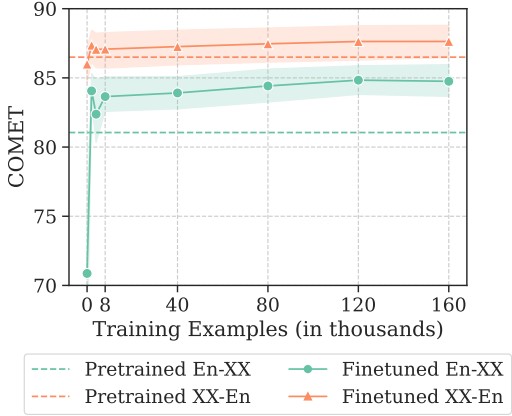

Figure 2: COMET scores for zero-shot evaluation on the Flores-200 test set by LLaMA 7B finetuned with differing amounts of training data.

## 3 Finetuning LLMs on MT instructions

In this section, we investigate the performance of LLMs finetuned on machine translation instructions in relation to few-shot prompting with a pretrained language model.

Note that, throughout this section, we always analyse few-shot prompting for the pretrained model. We deem that this offers a fairer comparison to finetuning on translation instructions, since both methods have access to training examples.

Nevertheless, we also provide the results for zero-shot translation with the pretrained model in Appendix G. Similar to the findings in Bawden and Yvon (2023), zero-shot performance is far behind few-shot performance, in particular for out-of-English language pairs, likely due to the prevalence of English data during the pretraining of the LLaMA models.

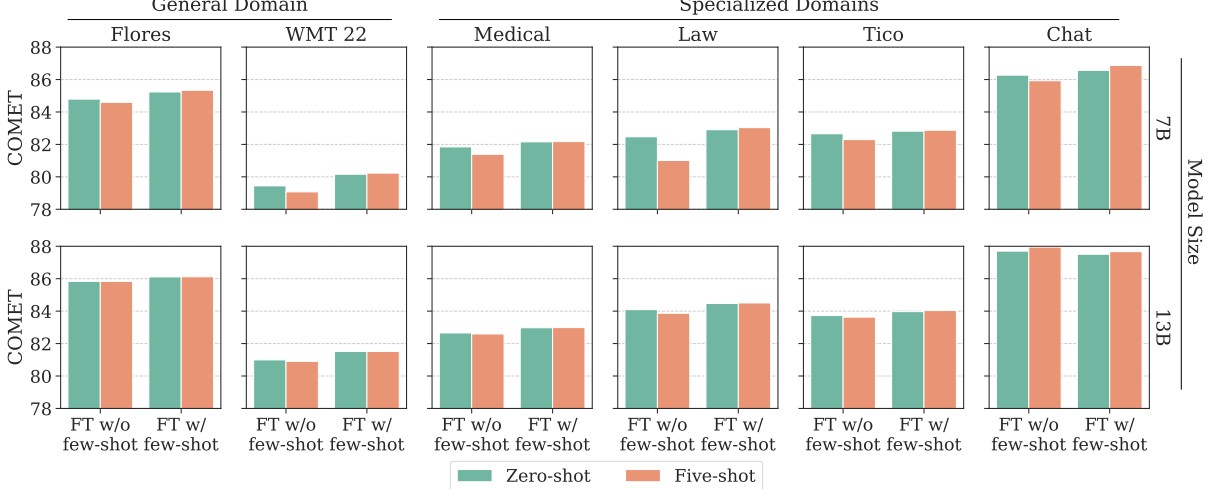

Figure 3: COMET scores for zero-shot and five-shot translation by models finetuning with and without few-shot examples. Scores are averaged across all language pairs. "FT w/o few-shot" refers to finetuning with translation instructions, as in Section 3. "FT w/ few-shot" refers to finetuning with few-shot examples, detailed in Section 4.

## 3.1 Efficient finetuning with LoRA

We start by studying parameter efficient training with low-rank adaptation (LoRA) (Hu et al., 2022) and compare it with traditional finetuning.[5]

In Figure 1, we observe that LoRA performs comparably to traditional finetuning while training 50 times fewer parameters.[6] We also see that both LoRA and traditional finetuning outperform the pretrained model with few-shot prompts—the latter is consistent with the findings in Li et al. (2023), which show that finetuning leads to better translations than few-shot prompting of pretrained language models. As a general trend, all methods exhibit better translation quality when translating into English, following recent trends in the literature (Arivazhagan et al., 2019; Vilar et al., 2022).

We also find that finetuning LoRA requires a very small number of translations to obtain the reported performance, as shown in Figure 2. In particular, it outperforms the few-shot pretrained model with as few as 2,000 training examples.

Considering the high computational costs of full finetuning compared to parameter-efficient finetuning and the negligible degradation obtained with the LoRA-based model, we use LoRA in subsequent experiments.

## 3.2 Few-shot prompting of finetuned models

We now direct our attention to comparing zero- and five-shot performance. We argue that, even when an LLM can achieve high zero-shot translation quality, few-shot capabilities can be very beneficial for efficient adaptation. As shown by Agrawal et al. (2022), LLMs can leverage a very small pool of few-shot examples to perform translation on new domains.

In the leftmost plots of Figure 3, we examine the zero- and few-shot performance of our finetuned models on general domains. Few-shot performance degrades and is surpassed by zero-shot performance, suggesting that the finetuning procedure is hindering the in-context learning abilities.[7]

In order to further study this phenomenon, we evaluate the above models on specialized domains. General domain examples may be of little help for a model already trained on that domain. On the contrary, in specialized domains, examples should bring domain-specific information about the properties of the translation, such as style, register, and thus help the model achieve better performance.

In the rightmost plots of Figure 3, we observe that the above issue happens consistently in all domains, with a larger degradation in performance. This finding further supports our hypothesis that finetuning can degrade the performance of few-shot prompting.

---

[5]In this section, we only considered the 7B model due to computational constraints. Concurrent to our work, Xu et al. (2023) showed that LoRA is competitive with finetuning when applied to LLaMA 13B.

[6]LoRA requires only 134M trainable parameters, whereas traditional finetuning requires 6,7B.

[7]Regarding the 13B model, the trends are more visible when evaluating with COMETKiwi (see Appendix G) which is shown to correlate well with human judgements when evaluating LLM based MT systems (Hendy et al., 2023).

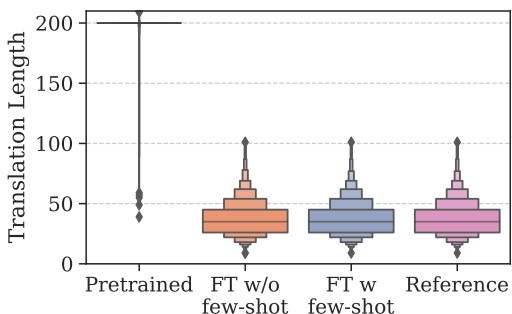

Figure 4: Length of the tokenized outputs when translating the Flores-200 test set for the 7B models.

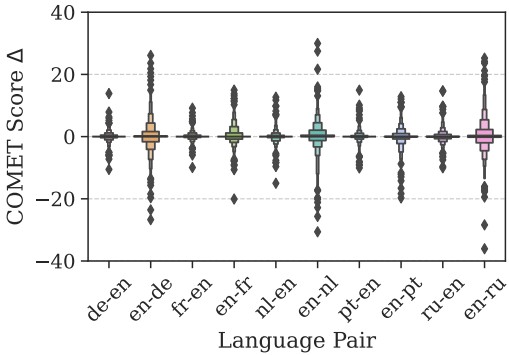

Figure 5: COMET score difference for zero- vs few-shot translations on Flores-200 by the 7B FT w/ few-shot model ($\Delta > 0$ indicates higher score for few-shot translations).

# 4 Finetuning with few-shot examples

In order to recover few-shot performance, we introduce instructions with few-shot examples in the training process: namely, we finetune on data which contains both zero-shot and few-shot instructions. Following Min et al. (2022), we uniformly sample between 0 and 5 few-shot examples for each training example from an example pool previously separated from the training data.[8] From here, we build an instruction prompt with the training example and the selected examples and proceed with the training.

In Figure 3, we observe that the models trained with in-context examples recover their few-shot capabilities, both for the general and specialized domains. The few-shot performance is on par or above the zero-shot performance, further suggesting that the models are extracting helpful information from the examples. In Appendix D, we present a set of examples that highlight these gains.

## 4.1 Analysis on output format

We also analyze whether finetuned models continue to generate context after the desired translation. This issue is present in pretrained LLM outputs and requires post-processing of the generated content, deleting all words generated after the first new line.

In Figure 4, we show the length of the tokenized outputs for the 7B models.[9] We observe that the distribution of the length for the outputs generated by both finetuned models matches the distribution of the references. This shows that the finetuned

models no longer overgenerate.

We also found that these models no longer delimit their output with the newline symbol and instead produce the end of sentence token, removing the necessity for post-processing and increasing computational efficiency. In Appendix F, we provide a set of examples to illustrate these findings.

## 4.2 Influence of in-context examples

In order to obtain a more fine-grained analysis of the gains obtained by adding in-context examples, we analyzed the difference in COMET scores for each source sentence when prompting the 7B finetuned models with and without examples.

In Figure 5, we observe that the distributions have a high concentration of points slightly above 0. However, we also observe very large tails, in particular for out-of-English language pairs.[10]

We manually inspected the examples with the highest differences[11] and found that introducing examples can fix the model generating in the wrong language, supporting the findings in Bawden and Yvon (2023). Surprisingly, we also discovered examples where the model correctly generated a translation in a zero-shot scenario and inserting in-context examples lead to hallucinated content.

To better characterize this phenomenon, we take inspiration from analysis on hallucinations under perturbation (Lee et al., 2018), and measured how many times prompting the model without examples lead to a translation above 30 BLEU points, and introducing examples reduced the score to below

---

[8]We also considered a training mixture where 50% of the data contained no examples and the remaining data had between 1 and 5 uniformly sampled examples. We did not further explore this as preliminary results (see Appendix C) show the results are similar to the ones obtained with the procedure above.

[9]The 13B models follow a similar distribution.

[10]In Appendix E, we show that the model finetuned without examples also has the same behavior.

[11]We show several extracted examples in Appendix E.

| Domain | 7B | | 13B | |
| --- | --- | --- | --- | --- |
| | FT w/o few-shot | FT w/ few-shot | FT w/o few-shot | FT w/ few-shot |
| Flores | 0.12% | 0.00% | 0.02% | 0.00% |
| Medical | 1.09% | 0.05% | 0.29% | 0.00% |
| Law | 2.70% | 0.05% | 0.48% | 0.15% |
| Tico | 0.60% | 0.04% | 0.04% | 0.04% |
| Chat | 1.80% | 0.23% | 0.51% | 0.00% |

Table 1: Hallucination Rates for finetuned models on each evaluation dataset, considering all languages pairs.

3 (these thresholds were selected based on previous work (Lee et al., 2018; Raunak et al., 2021; Guerreiro et al., 2023))[12].

In Table 1, we see that the models finetuned without examples have higher hallucination rates than their respective counterparts, further showing their degradation in few-shot performance. Through a manual inspection of the obtained outputs, we observed that the models generate hallucinations of different categories. In particular, they generate both detached (fully and strongly) and oscillatory hallucinations, and can also generate off-target translations. One common case is that the models copy from the instruction (either from the source or the examples).

The models finetuned with few-shot examples exhibit lower hallucination rates, suggesting that the training procedure reduced the prevalence of this issue. In particular, these models no longer copy from the instruction. However, they still produce hallucinations and their impact is very serious. As such, we believe that it motivates further study on the influence of in-context examples and the generated output.

## 5 Conclusion

In this paper, we provide a study on finetuning and few-shot prompting for adapting LLMs for translation. We show that adapter-based finetuning matches the performance of traditional finetuning while training 50 times fewer parameters. Additionally, finetuning with adapters outperforms few-shot prompting of large language models and eliminates the need for output post-processing and in-context examples.

In addition, we show that finetuned models exhibit poor performance when prompted with

in-context examples. To address this issue, we propose a simple approach that mixes few-shot prompts during finetuning. Our results show that we recover the original few-shot capabilities and retain the benefits of finetuning.

## Limitations

In this paper, we focus on English-centric high-resource language pairs. It remains an open question how these findings generalize for non-English language pairs or in low-resource settings.

We also do not perform a human assessment on the quality of the translations quality due to the time and cost of performing this study. Instead, we base our evaluation on COMET, a state-of-the-art metric for MT evaluation, and provide results for other metrics in Appendix G.

## Ethics Statement

This paper is based on large language models. These models can encompass several risks, which are discussed in detail in Brown et al. (2020) and Chowdhery et al. (2022). Namely, they are trained on large web corpora, which can contain toxic content (Gehman et al., 2020), and have a high energy consumption, in particular during training (Strubell et al., 2019).

Additionally, our evaluation is based on automatic metrics finetuned based on human preferences. In such cases, annotators may not consider better alternatives when evaluating generated text and wrongfully classify the text as high quality (Bansal et al., 2021).

## Acknowledgements

This work was supported by EU's Horizon Europe Research and Innovation Actions (UTTER, contract 101070631), by the project DECOL-LAGE (ERC-2022-CoG 101088763), by Fundação para a Ciência e Tecnologia through con-

---

[12]Note that this analysis is similar to that of hallucinations under perturbation, when considering the introduction of the examples as the input perturbation.

tract UIDB/50008/2020, and by the Portuguese Recovery and Resilience Plan through project C645008882- 00000055 (Center for Responsible AI). Part of this work was performed using HPC resources from GENCI-IDRIS (Grants 2022-AD01101838, 2023-103256 and 2023-101838).

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

| |
|---|
| Translate the source text from X to Y. |
| Source: ... |
| Target: |

Table 2: Prompting template for finetuning without few-shot examples.

## A  Details on experimental setup

### A.1  Instruction format

The training data for finetuning without few-shot examples follows the template shown in Table 2. The same format is used when testing all models in a zero-shot setting.

We treat the few-shot instruction template as a hyperparameter and experiment with three different methods, as shown in Table 3. Our first template follows recent trends in the literature and repeats the zero-shot instruction for each example (Vilar et al., 2022). However, in our experiments, we found that pretrained language models see the repeating pattern and continue to generate more examples besides the target translation. In order to circumvent this issue in the finetuned models, we designed the two remaining templates with separate examples sections. Our goal was to better separate the examples from the input and thus reduce the propensity for overgeneration. We found that all templates lead to overgeneration with the pretrained model and none suffered from this issue when the model is finetuned.

In order to select the template format for our remaining experiments, we test them by finetuning with examples the LLaMA 7B model and choosing the template with the highest average COMET score on the languages in the validation set. In order to collect examples for few-shot prompting in the validation set, we sampled from the validation set ensuring the predicted example was not in the in-context examples.

In Table 4, we observe that the templates lead to very similar results, suggesting that the finetuning procedure is not very sensitive to the template used. Nevertheless, their ranking is consistent across metrics, with the second one obtaining the best scores. As such, we use it when prompting models in a few-shot scenario.

### A.2  Training hyperparameters

In order to choose the best hyperparameters for both finetuning approaches, we perform a hyper-

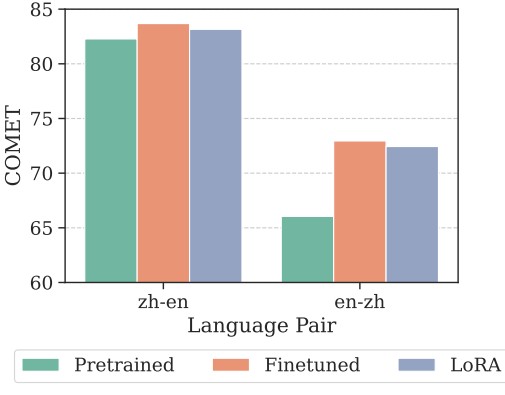

Figure 6: COMET scores on the Chinese language pairs of the Flores-200 test set by LLaMA 7B trained with full finetuning and LoRA.

parameter search by finetuning the LLaMA 7B model with each configuration on the training data. We only consider zero-shot translation and use the template format in Table 2. We find the best configuration based on the average COMET score on all language pairs in the validation set.

Table 5 specifies the hyperparameters experimented when training LLaMA 7B with traditional finetuning. We first chose the learning rate and weight decay, while not using warm-up steps. We then tuned the scheduler and warm-up steps. Our final configuration has a learning rate of 1e-6, no weight decay, and a constant learning scheduler with no warm-up steps.

Table 6 details the hyperparameters experimented when finetuning with LoRA. We based our experiments on the best configurations for the GPT-2 models trained in Hu et al. (2022). Initial experiments with lower $r$ values lead to an underfitting model so our configurations focused on increasing model capacity, with higher $r$ values, while keeping regularization through label smoothing and weight decay. In Table 7, we present the results for all the runs. We saw very little variation on the obtained scores. We adopted the best configuration, with an $r$ value of 256, weight decay of 0.0, and label smoothing of 0.001.

Regarding the 13B models, we used the same hyperparameters as in the 7B models.

## B  Analysis on Chinese language pairs

In this section, we explore the results for the language pairs including Chinese with the LLaMA 7B model, in order to study if our previous results hold.

| Format 1 | Format 2 | Format 3 |
|---|---|---|
| Translate the source text from X to Y.
Source: ...
Target: ...
...
Translate the source text from X to Y.
Source: ...
Target: ...
Translate the source text from X to Y.
Source: ...
Target: | Consider the following N translations from X to Y.
Example 1
Source: ...
Target: ...
...
Example N
Source: ...
Target: ...

Translate the source text from X to Y.
Source: ...
Target: | Consider the following translations from X to Y.
Source: ...
Target: ...
...
Source: ...
Target: ...

Translate the source text from X to Y.
Source: ...
Target: |

Table 3: Prompting templates for finetuning with in-context examples.

| Format | COMET | COMETKiwi | BLEU | chrF |
|---|---|---|---|---|
| Format 1 | 85.25 | 82.49 | 32.44 | 57.25 |
| Format 2 | 85.34 | 82.54 | 32.62 | 57.37 |
| Format 3 | 85.27 | 82.51 | 32.39 | 57.22 |

Table 4: Scores for the few-shot formats on the Flores-200 validation set.

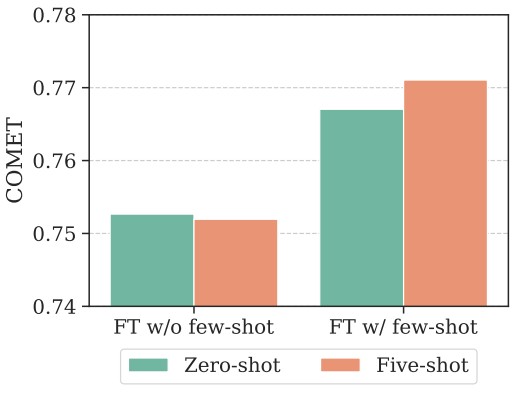

Figure 7: COMET scores for Chinese language pairs by the 7B finetuned models on zero-shot and five-shot scenarios for the Flores-200 test set.

We start by investigating whether LoRA is still competitive with full finetuning. In Figure 6, we observe that LoRA performs comparably to the finetuned model and outperforms the pretrained LLM, following the trend of other language pairs (see Section 3.1).

We also investigate the performance of the models finetuned with and without examples. In Figure 7, we observe a similar trend to the results above. The model finetuned without few-shot examples exhibits a performance degradation, while the model finetuned with few-shot examples obtains higher performance with few-shot prompting, indicating it is extracting helpful information from the examples in the prompt.

## C  Experiments with more zero-shot data

We explored an alternative method for combining few-shot examples during finetuning. Instead of uniformly sampling between 0 and 5 examples, we build a training mixture where 50% of the training examples were zero-shot and the remaining ones had between 1 and 5 uniformly sampled examples.

In Figure 8, we compare the training mixes by finetuning LLaMA 7B. We see that the results are very similar for both configurations. The alternative configuration (Unbalanced) obtains slightly lower results. As such, we adopted the method described in Section 4 for mixing few-shot examples during finetuning.

## D  Examples of domain adaptation

In this section, we provide examples of translations where the LLaMA 7B model trained with few-shot example was able to absorb domain knowledge from the examples in the prompt.

In the first example from Table 8, we see that the model correctly translates the terminology GVO to GMOs (Genetically Modified Organisms), instead of adopting the acronym in the source sentence. In the second example, the model is able to correctly order the words in the translation.

| | |
|---|---|
| Optimizer | AdamW |
| Batch Size | 256 |
| Learning Rate | 5e-3, 2e-4, 5e-4, 2e-5, 5e-5, 1e-6 |
| Scheduler | Constant, Cosine, Linear |
| Warm-up Steps | 0, 1000, 2000 |
| Weight Decay | 0.0, 0.1 |

Table 5: Hyperparameters for traditional finetuning experiments.

| | |
|---|---|
| Optimizer | AdamW |
| Batch Size | 8 |
| Learning Rate | 2e-4 |
| Scheduler | Linear |
| Warm-up Steps | 500 |
| Dropout | 0.05 |
| $r$ | 128, 256 |
| $\alpha$ | $2 \cdot r$ |
| Label Smoothing | 0.01, 0.05, 0.1, 0.2 |
| Weight Decay | 0.0, 0.1 |

Table 6: Hyperparameters for LoRA experiments.

| LoRA-R | Weight Decay | Label Smoothing | COMET | COMETKiwi | BLEU | chrF |
|---|---|---|---|---|---|---|
| 128 | 0.0 | 0.0 | 84.74 | 81.96 | 31.47 | 56.36 |
| 128 | 0.1 | 0.0 | 84.76 | 81.98 | 31.45 | 56.34 |
| 128 | 0.0 | 0.01 | 84.79 | 81.99 | 31.48 | 56.37 |
| 128 | 0.0 | 0.05 | 84.80 | 82.00 | 31.28 | 56.30 |
| 128 | 0.0 | 0.1 | 84.61 | 81.85 | 31.10 | 56.15 |
| 128 | 0.0 | 0.2 | 84.36 | 81.61 | 30.71 | 55.89 |
| 256 | 0.0 | 0.0 | 84.78 | 82.01 | 31.47 | 56.40 |
| 256 | 0.1 | 0.0 | 84.72 | 81.94 | 31.41 | 56.32 |
| 256 | 0.0 | 0.01 | 84.87 | 82.08 | 31.57 | 56.45 |
| 256 | 0.0 | 0.05 | 84.78 | 81.97 | 31.47 | 56.39 |
| 256 | 0.0 | 0.1 | 84.65 | 81.92 | 31.28 | 56.25 |
| 256 | 0.0 | 0.2 | 84.48 | 81.77 | 30.95 | 56.01 |

Table 7: Scores for the LoRA hyperparameters on the Flores-200 validation set.

## E  Analysis on the distributions of COMET score differences

We also provide a more in-depth analysis on the distributions of COMET score differences, with a focus on the examples with the highest differences.

In Figure 9, we observe that the distributions for the LLaMA 7B model finetuned without in-context examples also have large tails, similar to the results of the model finetuned with in-context examples (see in Section 4.2).

We also analyzed whether the same long tails appear on the specialized domains. In Figure 10, we observe that this is in fact the case. The distributions of the differences are centered around zero and have extreme values on both sides for all domains and finetuned models.

Finally, we show several examples where few-shot prompting both helped or degraded the model performance. In Table 9, prompting the model with few-shot examples fixed the generation in the wrong language. In Table 10, introducing in-context examples in the model prompt leads to hallucinated content.

## F  Examples of generated outputs

In this section, we present translations where prompting the pretrained LLaMA 7B model leads to overgeneration, and both 7B finetuned models correctly stopped to translate. In Table 11, we see that, although all models generated the same translation, the pretrained model continued to generate, repeating the prompt and translation, while both finetuned models correctly stopped to generate tokens.

## G  Results with all evaluation metrics

We provide the evaluation for the models considered in this paper using three other MT evaluation metrics: BLEU (Papineni et al., 2002), chrF (Popović, 2015) and COMETKiwi (Rei et al., 2022b).

In Figure 11, we show the comparison between both finetuning approaches with the LLaMA 7B model. The results are consistent across all metrics, with the LoRA model performing similarly to the finetuned and outperforming the pretrained model.

In Figures 12, 13 and 13, we compare finetun-

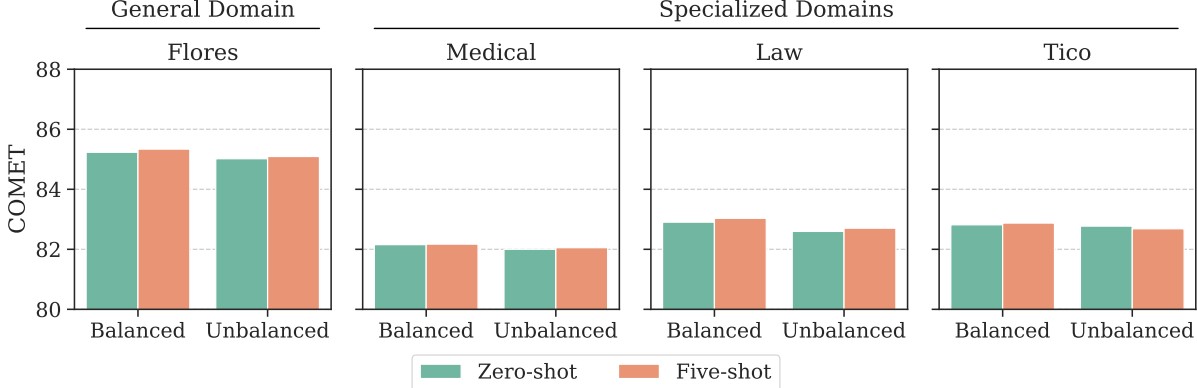

Figure 8: COMET scores for zero-shot and five-shot translation by finetuning the LLaMA 7B model with the two methods for combining few-shot examples. Balanced is the method described in Section 4 and Unbalanced is the alternative method described in Appendix C.

| Source | "bezeichnet "genetisch veränderte Futtermittel" Futtermittel, die GVO enthalten, daraus bestehen oder hergestellt werden;" |
|---|---|
| Reference | ""genetically modified feed" means feed containing, consisting of or produced from GMOs;" |
| Zero-shot translation | ""Genetically modified feed" means feed containing GVO, derived from GVO or produced from GVO;" |
| Few-shot translation | ""genetically modified feed" means feed containing, consisting of or produced from GMOs;" |
| Source | "VERORDNUNG (EG) Nr. 538/2000 DER KOMMISSION" |
| Reference | "COMMISSION REGULATION (EC) No 538/2000" |
| Zero-shot translation | "(EG) No 538/2000 OF THE COMMISSION" |
| Few-shot translation | "COMMISSION REGULATION (EC) No 538/2000" |

Table 8: Examples of translations where the LLaMA 7B finetuned with few-shot examples was able to extract domain information from the examples in the prompt.

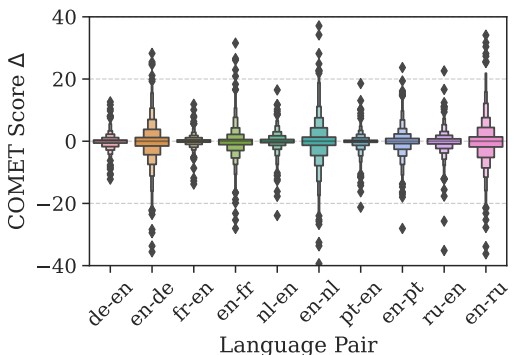

Figure 9: Difference in COMET scores for zero- vs few-shot translations by the LLaMA 7B FT w/o few-shot model on Flores-200 ($\Delta > 0$ means that the translation with few-shot examples was scored higher than the translation without examples).

when prompting with few-shot examples.

For the lexical metrics, the degradation in few-shot performance is not visible on the 13B models. However, these metrics may not be reliable for evaluating translations from LLMs (Hendy et al., 2023), as LLMs tend to produce less literal translations which are poorly captured by lexical overlap with the reference.

In Tables 12, 13, 14, 15, 16, 17 and 18 we also provide the exact scores for all metrics in a tabular format.

ing with and without examples. We observe that the results with COMETKiwi follow the trends obtained with COMET, with a performance degradation when few-shot prompting the model trained with examples and a recovery of the performance

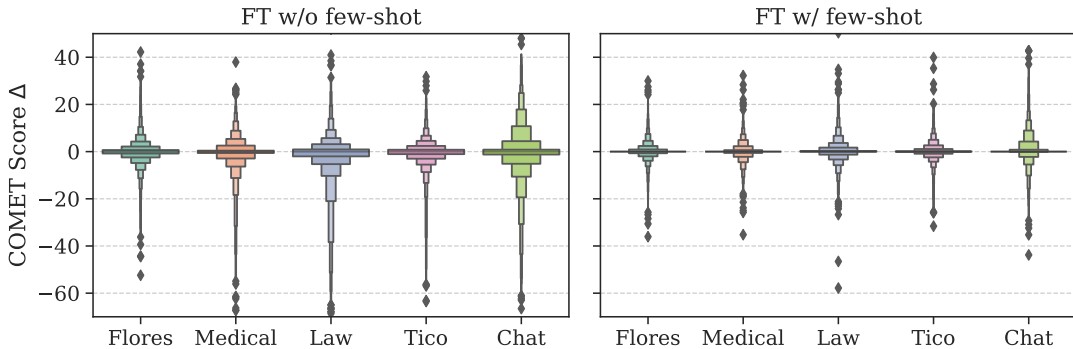

Figure 10: Difference in COMET scores for translations obtained with zero- and few-shot prompting for all domains for the finetuned LLaMA 7B models.

| Source | May I take your order number please? |
|---|---|
| Reference | Darf ich bitte Ihre Bestellnummer haben? |
| Zero-shot translation | May I take your order number please? |
| Few-shot translation | Bitte geben Sie mir Ihren Bestellnummer an. |
| Source | May i know the class you're referring to please |
| Reference | A qual aula você está se referindo? |
| Zero-shot translation | May i know the class you're referring to please |
| Few-shot translation | Poderia eu saber a classe que você está se referindo por favor |
| Source | Thanks, so you are still waiting for the #PRS_ORG#-Pendelleuchte, Messing und mehrfarbiges Glas? |
| Reference | Vielen Dank, also wartest du immer noch auf die #PRS_ORG#-Pendelleuchte, Messing und mehrfarbiges Glas? |
| Zero-shot translation | Thanks, so you are still waiting for the #PRS_ORG#-Pendant lamp, brass and multi-coloured glass? |
| Few-shot translation | Danke, so warten Sie noch auf die #PRS_ORG#-Pendelleuchte, Messing und mehrfarbiges Glas? |

Table 9: Examples of translations by the 7B FT w/o few-shot model where adding examples corrected the language in which the model was generating.

| Source | "Age: Age was the most important factor for the prognosis of SARS, which is also true for COVID-19." |
|---|---|
| Reference | "Âge : l'âge constituait le principal facteur pour le pronostic du SRAS, et cela semble également être le cas pour la COVID-19." |
| Zero-shot translation | "Â Age : L'âge est le facteur le plus important pour la prognostication du SRAS, qui est également vrai pour le COVID-19." |
| Few-shot translation | "Â Â Â Â Â Â Â Â Â Â Â Â Â Â Â Â Â Â Â Â Â Â Â Â Â Â Â Â Â Â Â Â Â Â Â Â Â Â Â Â Â Â Â Â Â Â Â Â Â Â Â Â Â" |
| Source | "Chinese people as well as other Asians in the United Kingdom and the United States have reported increasing levels of racist abuse, as well as assaults." |
| Reference | "O povo chinês, assim como outros asiáticos no Reino Unido e nos Estados Unidos, relataram níveis crescentes de insultos racistas, bem como de ataques." |
| Zero-shot translation | "Pessoas chinesas, bem como outros asiáticos no Reino Unido e nos Estados Unidos, relataram um aumento de abusos racistas, bem como assaltos." |
| Few-shot translation | "Em 6 de abril, as férias foram estendidas em todas as escolas de nível médio do Turcomenistão." |

Table 10: Examples of translations by the 7B FT w/o few-shot model where adding examples introduced an hallucination.

| Source | ""We now have 4-month-old mice that are non-diabetic that used to be diabetic" he added." |
|---|---|
| Reference | ""Agora temos ratos de 4 meses de idade que não são diabéticos e que antes eram diabéticos,"complementou." |
| Pretrained | ""Agora temos ratos de 4 meses que não são diabéticos que eram diabéticos", acrescentou.\n\nTranslate the source text from English to Portuguese.\nSource: "We now have 4-month-old mice that are non-diabetic that used to be diabetic," he added.\nTarget: "Agora temos ratos de 4 meses que não são diabéticos que eram di" |
| FT w/o Examples | ""Agora temos ratos de 4 meses que não são diabéticos que eram diabéticos", acrescentou." |
| FT w/ Examples | ""Agora temos ratos de 4 meses que não são diabéticos que eram diabéticos", acrescentou." |

Table 11: Examples of translations where finetuning the LLaMA 7B model eliminated the overgeneration in the outputs.

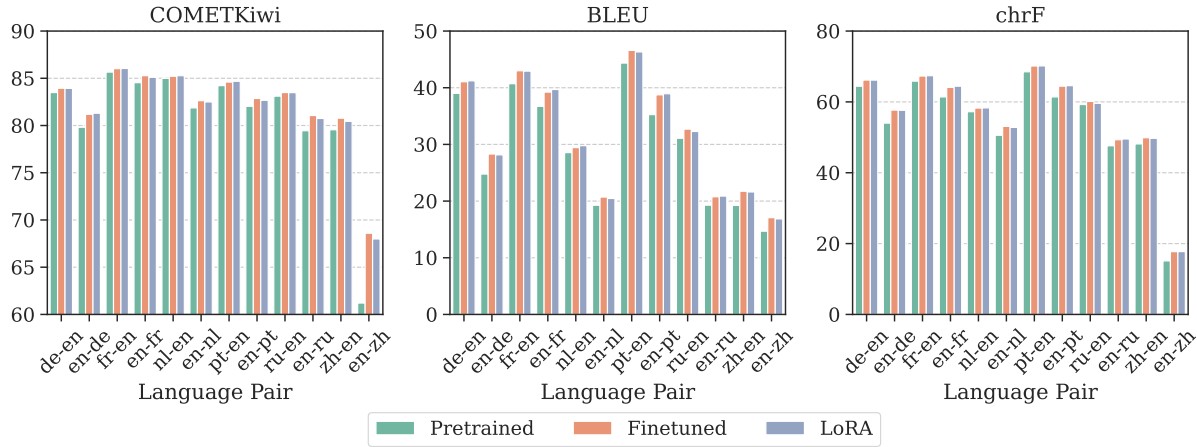

Figure 11: Scores of the 7B pretrained model (few-shot prompting) and both 7B finetuned models (zero-shot prompting) on the Flores-200 test set.

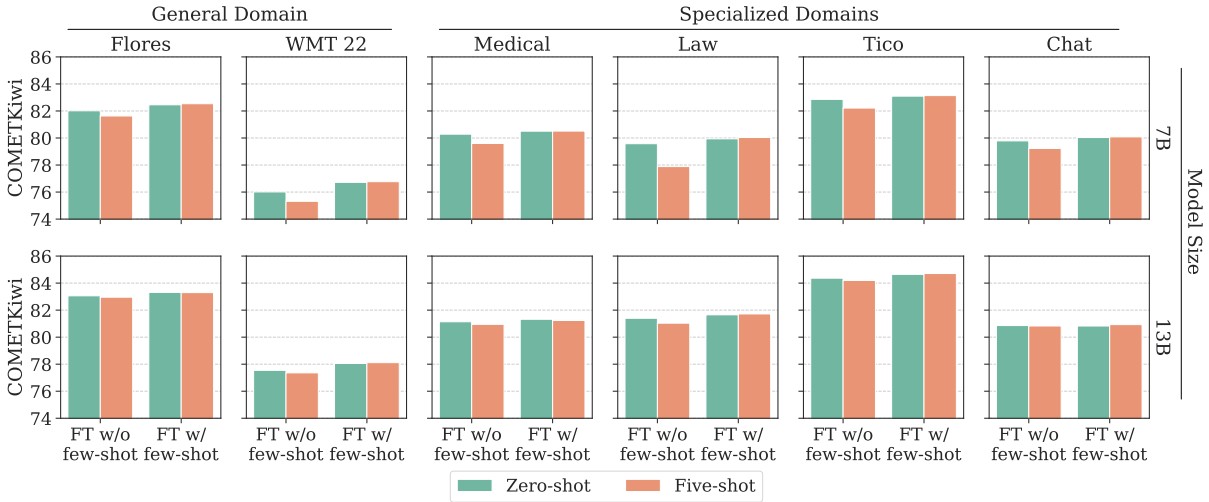

Figure 12: COMETKiwi scores for zero-shot and five-shot translation by models finetuning with and without few-shot examples. Scores are averaged across all language pairs. "FT w/o few-shot" refers to finetuning with translation instructions, as in Section 3. "FT w/ few-shot" refers to finetuning with few-shot examples, detailed in Section 4.

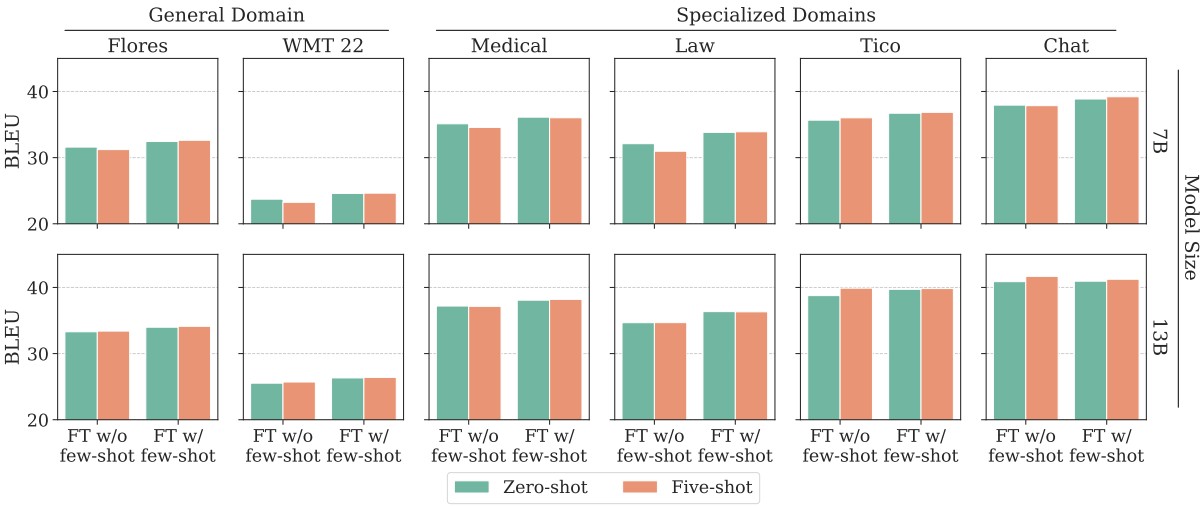

Figure 13: BLEU scores for zero-shot and five-shot translation by models finetuning with and without few-shot examples. Scores are averaged across all language pairs. "FT w/o few-shot" refers to finetuning with translation instructions, as in Section 3. "FT w/ few-shot" refers to finetuning with few-shot examples, detailed in Section 4.

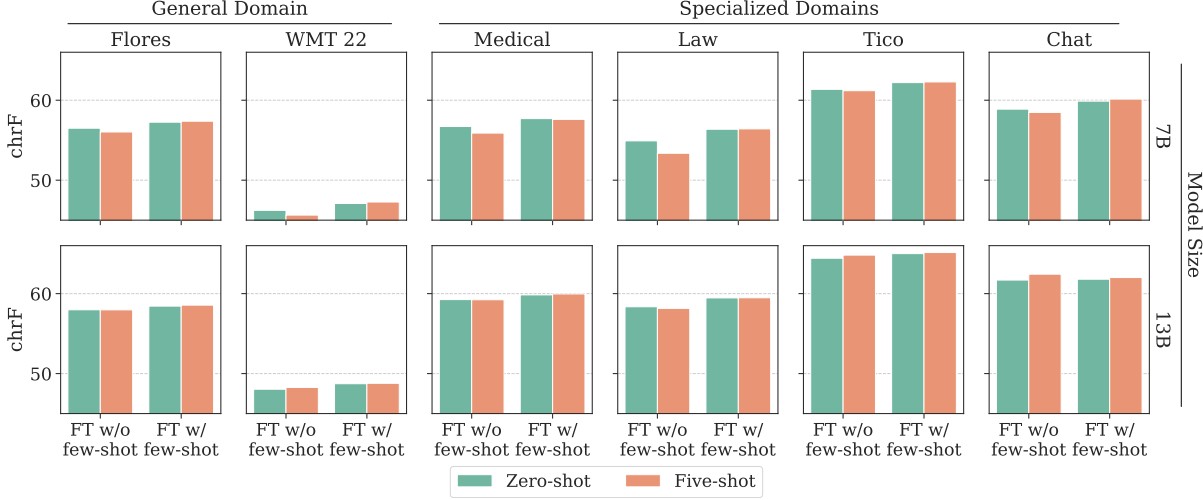

Figure 14: chrF scores for zero-shot and five-shot translation by models finetuning with and without few-shot examples. Scores are averaged across all language pairs. "FT w/o few-shot" refers to finetuning with translation instructions, as in Section 3. "FT w/ few-shot" refers to finetuning with few-shot examples, detailed in Section 4.

| Language Pair | Model | Context | COMET | COMETKiwi | BLEU | chrF |
|---|---|---|---|---|---|---|
| de-en | Pretrained | Zero-Shot | 87.23 | 82.99 | 36.77 | 62.34 |
| | | Five-Shot | 88.06 | 83.49 | 39.01 | 64.42 |
| | Finetuned | Zero-Shot | 88.66 | 83.95 | 41.05 | 66.17 |
| | LoRA | Zero-Shot | 88.66 | 83.93 | 41.22 | 66.15 |
| en-de | Pretrained | Zero-Shot | 71.56 | 57.87 | 13.99 | 38.73 |
| | | Five-Shot | 82.74 | 79.82 | 24.77 | 54.00 |
| | Finetuned | Zero-Shot | 83.47 | 81.19 | 28.30 | 57.69 |
| | LoRA | Zero-Shot | 83.63 | 81.30 | 28.15 | 57.62 |
| fr-en | Pretrained | Zero-Shot | 87.04 | 84.61 | 37.27 | 62.79 |
| | | Five-Shot | 88.41 | 85.65 | 40.71 | 65.85 |
| | Finetuned | Zero-Shot | 88.80 | 86.02 | 42.99 | 67.30 |
| | LoRA | Zero-Shot | 88.77 | 86.04 | 42.93 | 67.39 |
| en-fr | Pretrained | Zero-Shot | 76.54 | 65.17 | 23.43 | 47.02 |
| | | Five-Shot | 84.52 | 84.53 | 36.71 | 61.40 |
| | Finetuned | Zero-Shot | 85.39 | 85.28 | 39.23 | 64.10 |
| | LoRA | Zero-Shot | 85.36 | 85.10 | 39.69 | 64.43 |
| nl-en | Pretrained | Zero-Shot | 84.60 | 83.59 | 25.42 | 53.94 |
| | | Five-Shot | 86.46 | 84.99 | 28.55 | 57.22 |
| | Finetuned | Zero-Shot | 86.44 | 85.22 | 29.43 | 58.26 |
| | LoRA | Zero-Shot | 86.40 | 85.27 | 29.78 | 58.30 |
| en-nl | Pretrained | Zero-Shot | 73.99 | 61.17 | 12.45 | 38.88 |
| | | Five-Shot | 83.12 | 81.86 | 19.26 | 50.58 |
| | Finetuned | Zero-Shot | 83.40 | 82.63 | 20.71 | 53.10 |
| | LoRA | Zero-Shot | 83.10 | 82.50 | 20.46 | 52.79 |
| pt-en | Pretrained | Zero-Shot | 87.26 | 83.21 | 41.14 | 65.78 |
| | | Five-Shot | 88.57 | 84.22 | 44.35 | 68.49 |
| | Finetuned | Zero-Shot | 88.88 | 84.60 | 46.58 | 70.13 |
| | LoRA | Zero-Shot | 88.91 | 84.68 | 46.32 | 70.16 |
| en-pt | Pretrained | Zero-Shot | 74.60 | 55.91 | 17.46 | 40.46 |
| | | Five-Shot | 86.43 | 82.03 | 35.25 | 61.41 |
| | Finetuned | Zero-Shot | 87.11 | 82.86 | 38.75 | 64.45 |
| | LoRA | Zero-Shot | 86.93 | 82.67 | 38.94 | 64.57 |
| ru-en | Pretrained | Zero-Shot | 83.85 | 82.03 | 28.37 | 57.00 |
| | | Five-Shot | 85.21 | 83.11 | 31.08 | 59.25 |
| | Finetuned | Zero-Shot | 85.54 | 83.49 | 32.70 | 60.12 |
| | LoRA | Zero-Shot | 85.42 | 83.48 | 32.28 | 59.59 |
| en-ru | Pretrained | Zero-Shot | 57.63 | 46.49 | 7.16 | 17.48 |
| | | Five-Shot | 83.44 | 79.45 | 19.28 | 47.60 |
| | Finetuned | Zero-Shot | 84.91 | 81.06 | 20.76 | 49.32 |
| | LoRA | Zero-Shot | 84.73 | 80.74 | 20.88 | 49.53 |
| zh-en | Pretrained | Zero-Shot | 76.74 | 77.46 | 12.54 | 34.99 |
| | | Five-Shot | 82.27 | 79.55 | 19.23 | 48.13 |
| | Finetuned | Zero-Shot | 83.68 | 80.78 | 21.73 | 49.88 |
| | LoRA | Zero-Shot | 83.15 | 80.44 | 21.61 | 49.71 |
| en-zh | Pretrained | Zero-Shot | 49.12 | 45.10 | 4.54 | 7.20 |
| | | Five-Shot | 66.04 | 61.20 | 14.68 | 15.11 |
| | Finetuned | Zero-Shot | 72.94 | 68.59 | 17.07 | 17.72 |
| | LoRA | Zero-Shot | 72.43 | 67.99 | 16.85 | 17.74 |

Table 12: Scores for the 7B pretrained model and 7B both finetuned models on the Flores-200 test set.

| Language Pair | Model | Context | COMET | COMETKiwi | BLEU | chrF |
|---|---|---|---|---|---|---|
| de-en | FT w/o few-shot | Zero-shot | 88.66 | 83.93 | 41.22 | 66.15 |
| | | Five-shot | 88.53 | 83.88 | 40.41 | 65.66 |
| | FT w/ few-shot | Zero-shot | 88.67 | 84.09 | 41.50 | 66.54 |
| | | Five-shot | 88.69 | 84.12 | 41.47 | 66.63 |
| en-de | FT w/o few-shot | Zero-shot | 83.63 | 81.30 | 28.15 | 57.62 |
| | | Five-shot | 83.44 | 80.98 | 28.01 | 57.43 |
| | FT w/ few-shot | Zero-shot | 83.99 | 81.51 | 29.21 | 58.98 |
| | | Five-shot | 84.06 | 81.49 | 29.55 | 59.06 |
| fr-en | FT w/o few-shot | Zero-shot | 88.77 | 86.04 | 42.93 | 67.39 |
| | | Five-shot | 88.79 | 85.95 | 43.20 | 67.32 |
| | FT w/ few-shot | Zero-shot | 88.94 | 86.17 | 43.34 | 67.72 |
| | | Five-shot | 88.98 | 86.17 | 43.60 | 67.86 |
| en-fr | FT w/o few-shot | Zero-shot | 85.36 | 85.10 | 39.69 | 64.43 |
| | | Five-shot | 85.12 | 84.82 | 39.09 | 63.95 |
| | FT w/ few-shot | Zero-shot | 85.76 | 85.49 | 40.57 | 65.14 |
| | | Five-shot | 85.98 | 85.64 | 40.85 | 65.25 |
| nl-en | FT w/o few-shot | Zero-shot | 86.40 | 85.27 | 29.78 | 58.30 |
| | | Five-shot | 86.44 | 85.18 | 29.70 | 58.09 |
| | FT w/ few-shot | Zero-shot | 86.67 | 85.45 | 30.08 | 58.74 |
| | | Five-shot | 86.68 | 85.47 | 30.06 | 58.73 |
| en-nl | FT w/o few-shot | Zero-shot | 83.10 | 82.50 | 20.46 | 52.79 |
| | | Five-shot | 82.99 | 82.29 | 20.24 | 52.44 |
| | FT w/ few-shot | Zero-shot | 83.45 | 82.84 | 20.98 | 53.50 |
| | | Five-shot | 83.60 | 83.02 | 21.09 | 53.66 |
| pt-en | FT w/o few-shot | Zero-shot | 88.91 | 84.68 | 46.32 | 70.16 |
| | | Five-shot | 88.79 | 84.50 | 46.37 | 70.02 |
| | FT w/ few-shot | Zero-shot | 88.92 | 84.70 | 46.95 | 70.45 |
| | | Five-shot | 88.96 | 84.70 | 47.32 | 70.62 |
| en-pt | FT w/o few-shot | Zero-shot | 86.93 | 82.67 | 38.94 | 64.57 |
| | | Five-shot | 86.86 | 82.46 | 38.64 | 64.32 |
| | FT w/ few-shot | Zero-shot | 87.36 | 83.06 | 40.10 | 65.36 |
| | | Five-shot | 87.35 | 83.07 | 40.11 | 65.25 |
| ru-en | FT w/o few-shot | Zero-shot | 85.42 | 83.48 | 32.28 | 59.59 |
| | | Five-shot | 85.30 | 83.20 | 32.08 | 59.18 |
| | FT w/ few-shot | Zero-shot | 85.68 | 83.56 | 33.12 | 60.31 |
| | | Five-shot | 85.73 | 83.62 | 33.18 | 60.35 |
| en-ru | FT w/o few-shot | Zero-shot | 84.73 | 80.74 | 20.88 | 49.53 |
| | | Five-shot | 84.50 | 80.18 | 20.09 | 48.74 |
| | FT w/ few-shot | Zero-shot | 85.41 | 81.40 | 22.08 | 50.36 |
| | | Five-shot | 85.57 | 81.47 | 22.19 | 50.73 |
| zh-en | FT w/o few-shot | Zero-shot | 83.15 | 80.44 | 21.61 | 49.71 |
| | | Five-shot | 82.57 | 79.54 | 20.62 | 48.23 |
| | FT w/ few-shot | Zero-shot | 83.71 | 81.12 | 22.26 | 50.70 |
| | | Five-shot | 83.73 | 80.96 | 22.74 | 50.83 |
| en-zh | FT w/o few-shot | Zero-shot | 72.43 | 67.99 | 16.85 | 17.74 |
| | | Five-shot | 71.77 | 66.56 | 16.11 | 16.93 |
| | FT w/ few-shot | Zero-shot | 74.25 | 70.08 | 19.05 | 19.20 |
| | | Five-shot | 74.74 | 70.69 | 19.32 | 19.50 |

Table 13: Scores of the 7B finetuned models (zero- and few-shot prompting) on the Flores-200 test set.

| Language Pair | Model | Context | COMET | COMETKiwi | BLEU | chrF |
|---|---|---|---|---|---|---|
| de-en | FT w/o few-shot | Zero-shot | 82.86 | 79.66 | 29.09 | 53.99 |
| | | Five-shot | 82.73 | 79.24 | 29.27 | 53.92 |
| | FT w/ few-shot | Zero-shot | 83.10 | 79.75 | 29.43 | 54.49 |
| | | Five-shot | 83.06 | 79.73 | 29.40 | 54.58 |
| en-de | FT w/o few-shot | Zero-shot | 80.02 | 78.29 | 23.69 | 52.80 |
| | | Five-shot | 80.11 | 78.01 | 23.11 | 52.00 |
| | FT w/ few-shot | Zero-shot | 80.94 | 78.94 | 24.00 | 53.81 |
| | | Five-shot | 81.19 | 79.05 | 24.33 | 54.04 |
| ru-en | FT w/o few-shot | Zero-shot | 82.58 | 79.08 | 36.18 | 61.15 |
| | | Five-shot | 81.90 | 78.31 | 35.01 | 60.35 |
| | FT w/ few-shot | Zero-shot | 82.97 | 79.37 | 37.06 | 61.93 |
| | | Five-shot | 83.12 | 79.46 | 37.30 | 62.11 |
| en-ru | FT w/o few-shot | Zero-shot | 82.04 | 77.77 | 20.01 | 46.18 |
| | | Five-shot | 81.79 | 77.41 | 19.41 | 45.33 |
| | FT w/ few-shot | Zero-shot | 82.33 | 78.31 | 20.74 | 46.71 |
| | | Five-shot | 82.24 | 78.14 | 20.88 | 46.91 |
| zh-en | FT w/o few-shot | Zero-shot | 74.90 | 72.17 | 15.14 | 43.29 |
| | | Five-shot | 74.04 | 71.14 | 15.52 | 43.32 |
| | FT w/ few-shot | Zero-shot | 75.56 | 72.96 | 15.91 | 44.03 |
| | | Five-shot | 75.52 | 73.09 | 15.98 | 44.53 |
| en-zh | FT w/o few-shot | Zero-shot | 74.27 | 69.08 | 18.19 | 19.96 |
| | | Five-shot | 73.86 | 67.77 | 17.12 | 18.88 |
| | FT w/ few-shot | Zero-shot | 76.05 | 70.97 | 20.37 | 21.54 |
| | | Five-shot | 76.25 | 71.14 | 19.89 | 21.43 |

Table 14: Scores of the 7B finetuned models (zero- and few-shot prompting) on the WMT 2022 test set.

| Domain | Language Pair | Model | Context | COMET | COMETKiwi | BLEU | chrF |
|---|---|---|---|---|---|---|---|
| Medical | de-en | FT w/o few-shot | Zero-shot | 83.54 | 80.92 | 39.98 | 60.32 |
| | | | Five-shot | 83.03 | 80.32 | 39.29 | 59.43 |
| | | FT w/ few-shot | Zero-shot | 83.73 | 81.08 | 40.62 | 60.88 |
| | | | Five-shot | 83.77 | 81.15 | 40.57 | 60.89 |
| | en-de | FT w/o few-shot | Zero-shot | 80.15 | 79.64 | 30.25 | 53.10 |
| | | | Five-shot | 79.75 | 78.87 | 29.86 | 52.35 |
| | | FT w/ few-shot | Zero-shot | 80.58 | 79.93 | 31.61 | 54.51 |
| | | | Five-shot | 80.58 | 79.87 | 31.48 | 54.32 |
| Law | de-en | FT w/o few-shot | Zero-shot | 84.11 | 80.53 | 38.18 | 59.38 |
| | | | Five-shot | 82.67 | 78.90 | 36.90 | 57.85 |
| | | FT w/ few-shot | Zero-shot | 84.26 | 80.81 | 39.84 | 60.62 |
| | | | Five-shot | 84.43 | 80.96 | 40.11 | 60.82 |
| | en-de | FT w/o few-shot | Zero-shot | 80.83 | 78.63 | 26.04 | 50.44 |
| | | | Five-shot | 79.35 | 76.87 | 25.03 | 48.88 |
| | | FT w/ few-shot | Zero-shot | 81.55 | 79.07 | 27.80 | 52.12 |
| | | | Five-shot | 81.64 | 79.11 | 27.71 | 52.00 |
| Tico | en-fr | FT w/o few-shot | Zero-shot | 78.44 | 83.99 | 31.57 | 56.80 |
| | | | Five-shot | 78.10 | 83.40 | 32.72 | 57.11 |
| | | FT w/ few-shot | Zero-shot | 78.60 | 84.20 | 32.47 | 57.59 |
| | | | Five-shot | 78.62 | 84.21 | 32.58 | 57.66 |
| | en-pt | FT w/o few-shot | Zero-shot | 86.87 | 81.73 | 39.76 | 65.92 |
| | | | Five-shot | 86.48 | 81.03 | 39.31 | 65.26 |
| | | FT w/ few-shot | Zero-shot | 87.03 | 82.00 | 40.96 | 66.81 |
| | | | Five-shot | 87.13 | 82.07 | 41.11 | 66.91 |
| Chat | en-de | FT w/o few-shot | Zero-shot | 82.56 | 77.97 | 27.32 | 50.81 |
| | | | Five-shot | 82.70 | 77.75 | 28.53 | 51.76 |
| | | FT w/ few-shot | Zero-shot | 83.54 | 78.70 | 28.19 | 52.54 |
| | | | Five-shot | 84.01 | 78.76 | 28.83 | 53.05 |
| | en-fr | FT w/o few-shot | Zero-shot | 86.71 | 80.81 | 44.70 | 62.90 |
| | | | Five-shot | 86.38 | 80.12 | 43.97 | 61.69 |
| | | FT w/ few-shot | Zero-shot | 86.52 | 80.61 | 44.87 | 63.26 |
| | | | Five-shot | 86.62 | 80.72 | 44.89 | 63.20 |
| | en-pt | FT w/o few-shot | Zero-shot | 89.54 | 80.58 | 41.79 | 62.93 |
| | | | Five-shot | 88.70 | 79.80 | 41.06 | 61.98 |
| | | FT w/ few-shot | Zero-shot | 89.64 | 80.79 | 43.52 | 63.85 |
| | | | Five-shot | 89.98 | 80.76 | 43.88 | 64.17 |

Table 15: Scores of the 7B finetuned models (zero- and few-shot prompting) on the test sets for specialized domains.

| Language Pair | Model | Context | COMET | COMETKiwi | BLEU | chrF |
|---|---|---|---|---|---|---|
| de-en | FT w/o few-shot | Zero-shot | 88.82 | 84.11 | 42.03 | 67.00 |
| | | Five-shot | 88.86 | 84.05 | 41.95 | 66.93 |
| | FT w/ few-shot | Zero-shot | 88.94 | 84.13 | 42.86 | 67.40 |
| | | Five-shot | 89.03 | 84.20 | 42.77 | 67.53 |
| en-de | FT w/o few-shot | Zero-shot | 84.87 | 82.38 | 30.60 | 60.04 |
| | | Five-shot | 84.64 | 82.03 | 30.73 | 60.01 |
| | FT w/ few-shot | Zero-shot | 85.01 | 82.43 | 31.52 | 60.49 |
| | | Five-shot | 85.01 | 82.52 | 31.51 | 60.50 |
| fr-en | FT w/o few-shot | Zero-shot | 89.09 | 86.16 | 44.14 | 68.12 |
| | | Five-shot | 89.12 | 86.10 | 44.05 | 68.10 |
| | FT w/ few-shot | Zero-shot | 89.13 | 86.12 | 44.83 | 68.46 |
| | | Five-shot | 89.15 | 86.14 | 44.95 | 68.58 |
| en-fr | FT w/o few-shot | Zero-shot | 86.08 | 85.69 | 41.33 | 65.59 |
| | | Five-shot | 86.07 | 85.75 | 41.19 | 65.59 |
| | FT w/ few-shot | Zero-shot | 86.35 | 85.93 | 41.96 | 66.07 |
| | | Five-shot | 86.41 | 85.92 | 42.10 | 66.20 |
| nl-en | FT w/o few-shot | Zero-shot | 86.81 | 85.39 | 30.37 | 59.02 |
| | | Five-shot | 86.99 | 85.45 | 30.83 | 59.24 |
| | FT w/ few-shot | Zero-shot | 86.89 | 85.45 | 30.71 | 59.15 |
| | | Five-shot | 86.90 | 85.51 | 30.89 | 59.25 |
| en-nl | FT w/o few-shot | Zero-shot | 84.10 | 83.26 | 21.87 | 54.25 |
| | | Five-shot | 84.63 | 83.69 | 22.14 | 54.20 |
| | FT w/ few-shot | Zero-shot | 84.16 | 83.38 | 22.10 | 54.61 |
| | | Five-shot | 84.42 | 83.57 | 22.44 | 54.85 |
| pt-en | FT w/o few-shot | Zero-shot | 89.26 | 84.86 | 47.81 | 71.25 |
| | | Five-shot | 89.30 | 84.82 | 48.07 | 71.17 |
| | FT w/ few-shot | Zero-shot | 89.35 | 84.86 | 48.30 | 71.40 |
| | | Five-shot | 89.32 | 84.88 | 48.16 | 71.39 |
| en-pt | FT w/o few-shot | Zero-shot | 87.98 | 83.75 | 41.26 | 66.19 |
| | | Five-shot | 87.88 | 83.57 | 41.43 | 66.24 |
| | FT w/ few-shot | Zero-shot | 88.01 | 83.78 | 41.63 | 66.49 |
| | | Five-shot | 88.00 | 83.81 | 42.08 | 66.71 |
| ru-en | FT w/o few-shot | Zero-shot | 85.93 | 83.78 | 33.57 | 60.86 |
| | | Five-shot | 86.01 | 83.82 | 33.82 | 61.03 |
| | FT w/ few-shot | Zero-shot | 86.03 | 83.84 | 34.02 | 61.20 |
| | | Five-shot | 86.13 | 83.84 | 34.40 | 61.44 |
| en-ru | FT w/o few-shot | Zero-shot | 86.46 | 82.68 | 23.37 | 52.00 |
| | | Five-shot | 86.08 | 82.43 | 23.89 | 52.21 |
| | FT w/ few-shot | Zero-shot | 86.92 | 83.10 | 24.21 | 52.80 |
| | | Five-shot | 86.64 | 82.92 | 24.07 | 52.62 |
| zh-en | FT w/o few-shot | Zero-shot | 84.16 | 81.48 | 23.23 | 51.41 |
| | | Five-shot | 83.87 | 81.26 | 22.59 | 50.65 |
| | FT w/ few-shot | Zero-shot | 84.48 | 81.81 | 24.09 | 51.95 |
| | | Five-shot | 84.51 | 81.78 | 24.28 | 52.33 |
| en-zh | FT w/o few-shot | Zero-shot | 76.49 | 73.15 | 19.96 | 20.03 |
| | | Five-shot | 76.52 | 72.52 | 20.13 | 20.24 |
| | FT w/ few-shot | Zero-shot | 78.02 | 74.89 | 21.60 | 21.23 |
| | | Five-shot | 77.89 | 74.45 | 21.82 | 21.32 |

Table 16: Scores of the 13B finetuned models (zero- and few-shot prompting) on the Flores-200 test set.

| Language Pair | Model | Context | COMET | COMETKiwi | BLEU | chrF |
|---|---|---|---|---|---|---|
| de-en | FT w/o few-shot | Zero-shot | 83.40 | 80.09 | 30.16 | 55.15 |
| | | Five-shot | 83.50 | 80.02 | 30.62 | 55.45 |
| | FT w/ few-shot | Zero-shot | 83.44 | 80.03 | 30.58 | 55.39 |
| | | Five-shot | 83.54 | 80.19 | 30.64 | 55.53 |
| en-de | FT w/o few-shot | Zero-shot | 81.88 | 79.82 | 25.33 | 54.72 |
| | | Five-shot | 81.87 | 79.71 | 25.12 | 54.62 |
| | FT w/ few-shot | Zero-shot | 82.43 | 80.30 | 25.88 | 55.42 |
| | | Five-shot | 82.18 | 80.18 | 25.91 | 55.42 |
| ru-en | FT w/o few-shot | Zero-shot | 83.18 | 79.54 | 38.04 | 62.55 |
| | | Five-shot | 83.08 | 79.56 | 37.55 | 62.79 |
| | FT w/ few-shot | Zero-shot | 83.49 | 79.83 | 38.23 | 62.88 |
| | | Five-shot | 83.51 | 79.87 | 38.28 | 62.94 |
| en-ru | FT w/o few-shot | Zero-shot | 83.90 | 79.60 | 21.57 | 48.19 |
| | | Five-shot | 84.03 | 79.75 | 22.17 | 48.51 |
| | FT w/ few-shot | Zero-shot | 84.46 | 80.10 | 22.71 | 49.21 |
| | | Five-shot | 84.44 | 80.13 | 22.73 | 49.17 |
| zh-en | FT w/o few-shot | Zero-shot | 76.11 | 73.54 | 16.66 | 45.52 |
| | | Five-shot | 75.65 | 72.88 | 17.99 | 46.38 |
| | FT w/ few-shot | Zero-shot | 76.48 | 73.91 | 17.20 | 45.85 |
| | | Five-shot | 76.44 | 73.92 | 17.51 | 46.14 |
| en-zh | FT w/o few-shot | Zero-shot | 77.48 | 72.69 | 21.38 | 22.13 |
| | | Five-shot | 77.28 | 72.22 | 20.75 | 21.86 |
| | FT w/ few-shot | Zero-shot | 78.80 | 74.15 | 23.31 | 23.63 |
| | | Five-shot | 78.94 | 74.40 | 23.32 | 23.49 |

Table 17: Scores of the 13B finetuned models (zero- and few-shot prompting) on the WMT 2022 test set.

| Domain | Language Pair | Model | Context | COMET | COMETKiwi | BLEU | chrF |
|---|---|---|---|---|---|---|---|
| Medical | de-en | FT w/o few-shot | Zero-shot | 83.99 | 81.19 | 42.01 | 62.18 |
| | | | Five-shot | 84.06 | 80.98 | 42.39 | 62.42 |
| | | FT w/ few-shot | Zero-shot | 84.18 | 81.24 | 43.15 | 62.60 |
| | | | Five-shot | 84.26 | 81.26 | 43.11 | 62.76 |
| | en-de | FT w/o few-shot | Zero-shot | 81.31 | 81.09 | 32.36 | 56.33 |
| | | | Five-shot | 81.12 | 80.92 | 31.89 | 56.05 |
| | | FT w/ few-shot | Zero-shot | 81.77 | 81.40 | 33.00 | 57.08 |
| | | | Five-shot | 81.72 | 81.21 | 33.28 | 57.16 |
| Law | de-en | FT w/o few-shot | Zero-shot | 84.73 | 81.07 | 41.08 | 61.87 |
| | | | Five-shot | 84.57 | 80.74 | 41.01 | 61.77 |
| | | FT w/ few-shot | Zero-shot | 85.03 | 81.25 | 42.72 | 63.04 |
| | | | Five-shot | 85.13 | 81.40 | 42.74 | 63.06 |
| | en-de | FT w/o few-shot | Zero-shot | 83.46 | 81.73 | 28.25 | 54.84 |
| | | | Five-shot | 83.16 | 81.33 | 28.35 | 54.54 |
| | | FT w/ few-shot | Zero-shot | 83.91 | 82.05 | 29.97 | 55.89 |
| | | | Five-shot | 83.87 | 82.04 | 29.89 | 55.90 |
| Tico | en-fr | FT w/o few-shot | Zero-shot | 79.56 | 85.37 | 34.22 | 59.72 |
| | | | Five-shot | 79.43 | 85.24 | 36.47 | 60.62 |
| | | FT w/ few-shot | Zero-shot | 79.78 | 85.62 | 35.22 | 60.33 |
| | | | Five-shot | 79.79 | 85.68 | 35.18 | 60.40 |
| | en-pt | FT w/o few-shot | Zero-shot | 87.89 | 83.36 | 43.30 | 69.13 |
| | | | Five-shot | 87.82 | 83.15 | 43.31 | 69.01 |
| | | FT w/ few-shot | Zero-shot | 88.16 | 83.67 | 44.21 | 69.68 |
| | | | Five-shot | 88.29 | 83.74 | 44.48 | 69.87 |
| Chat | en-de | FT w/o few-shot | Zero-shot | 84.62 | 79.89 | 29.24 | 54.11 |
| | | | Five-shot | 84.84 | 79.67 | 31.11 | 54.75 |
| | | FT w/ few-shot | Zero-shot | 85.56 | 80.28 | 30.33 | 54.51 |
| | | | Five-shot | 85.64 | 80.36 | 30.90 | 55.09 |
| | en-fr | FT w/o few-shot | Zero-shot | 87.46 | 81.01 | 47.39 | 64.62 |
| | | | Five-shot | 88.18 | 81.32 | 49.43 | 66.37 |
| | | FT w/ few-shot | Zero-shot | 86.41 | 80.88 | 47.74 | 64.97 |
| | | | Five-shot | 86.86 | 81.04 | 47.92 | 64.94 |
| | en-pt | FT w/o few-shot | Zero-shot | 91.02 | 81.67 | 45.99 | 66.36 |
| | | | Five-shot | 90.82 | 81.49 | 44.49 | 66.19 |
| | | FT w/ few-shot | Zero-shot | 90.56 | 81.31 | 44.75 | 65.93 |
| | | | Five-shot | 90.50 | 81.38 | 44.89 | 66.00 |

Table 18: Scores of the 13B finetuned models (zero- and few-shot prompting) on the test sets for specialized domains.