# OpenReview forum: "Steering Large Language Models for Machine Translation with Finetuning and In-Context Learning"
_EMNLP/2023/Conference — EMNLP 2023 Findings_

### Official Review · Reviewer_Rre2 · 2023-08-03

**Typos Grammar Style And Presentation Improvements:** 1. In Figure 3, it's better to presen…
**Soundness:** 4

**Excitement:**

4: Strong: This paper deepens the understanding of some phenomenon or lowers the barriers to an existing research direction.

**Paper Topic And Main Contributions:**

This paper mainly discusses how to tune a pre-trained language model for machine translation. Firstly, as previous works, it finds that parameter-efficient fine-tuning offers comparable performance as full-finetuning while only tuning a negligible amount of parameters. Next, it explores the role of prompting during fine-tuning and finds out that fine-tuning with few-shot prompting offers better results for both in-domain and out-of-domain translation.

The main contributions of this paper are:
1. It combines both in-context tuning and fine-tuning to tune a large pre-trained language model, which is novel.
2. The main conclusions are supported by extensive experiments and analysis.

**Questions For The Authors:**

1. Could you explain Lines 517-519 in more detail? I don't get it.

2. Could you offer the results for different templates in Table 3?

3. For Table 4 and 5, could you also offer the results for different settings? I'm interested in the variance.

4. In Figure 2, do you have results for full fine-tuning?

**Reasons To Accept:**

1. The paper is easy to follow;
2. Extensive experiments are conducted to support the main findings;
3. The analysis is thorough.

**Reasons To Reject:**

1. The novelty of the method is limited, but I like the analysis;
2. Only one pre-trained model is applied. I would be interested in the analysis of different models, like different sizes of LLaMA, and FLAN-finetuned models.

**Reproducibility:**

3: Could reproduce the results with some difficulty. The settings of parameters are underspecified or subjectively determined; the training/evaluation data are not widely available.

**Reviewer Confidence:**

4: Quite sure. I tried to check the important points carefully. It's unlikely, though conceivable, that I missed something that should affect my ratings.

---

> ### Author Rebuttal · Authors · 2023-08-28
>
> Thank you for your review.
>
> > "Only one pre-trained model is applied. I would be interested in the analysis of different models, like different sizes of LLaMA, and FLAN-finetuned models."
>
> We agree that considering scaled-up LLaMa versions can strengthen our work. We provide a comprehensive answer to your concerns regarding this in the response to R2, who also shared the same concerns. We will make sure to include this analysis in the final version.
>
> > "Could you explain Lines 517-519 in more detail? I don't get it."
>
> Regarding lines 517-519, the first template repeats the translation instruction multiple times. When a pretrained model is prompted with it, it sees the repeating pattern and continues to generate more examples besides the target translation. With our remaining templates, we designed a separate examples section that would better separate the examples from the input. Our goal was to reduce the propensity for overgeneration. However, we found that all templates lead to overgeneration with the pretrained model and none suffer from this issue when the model is finetuned. In the camera ready version, we will better explain this information.
>
> > "Could you offer the results for different templates in Table 3?"
>
> The results for the different templates were very similar, as shown in the Table below. We believe that once the model learns which template to expect, it has little influence on the obtained results. Nevertheless, we would like to note that the ranking of the templates according to performance is very consistent across metrics.
>
> | Format | COMET | COMETKiwi | BLEU | chrF   |
> | -------: | ---------: | -------------: | ------: | ------: |
> | 1          | 85.25     | 82.49           | 32.44  | 57.25 |
> | 2          | 85.34     | 82.54           | 32.62  | 57.37 |
> | 3          | 85.27     | 82.51           | 32.39  | 57.22 |
>
> > "For Table 4 and 5, could you also offer the results for different settings? I'm interested in the variance."
>
> In the Table below, we present the average scores on the validation set for the hyperparameter configurations for the LoRA models. Note that we only tried label smoothing without weight decay, as our best model in initial experiments did not have weight decay. Overall, the results show a small variability across all the metrics.
>
> | LoRA-R | Weight Decay | Label Smoothing | COMET | COMETKiwi | BLEU | chrF   |
> | --------: | ---------------: | -------------------: | ---------: | -------------: | ------: | ------: |
> | 128       | 0.0                  | 0.0                        | 84.74     | 81.96            | 31.47 | 56.36 |
> | 128       | 0.1                  | 0.0                        | 84.76     | 81.98            | 31.45 | 56.34 |
> | 128       | 0.0                  | 0.01                      | 84.79     | 81.99            | 31.48 | 56.37 |
> | 128       | 0.0                  | 0.05                      | 84.80     | 82.00            | 31.28 | 56.30 |
> | 128       | 0.0                  | 0.1                        | 84.61     | 81.85            | 31.10 | 56.15 |
> | 128       | 0.0                  | 0.2                        | 84.36     | 81.61            | 30.71 | 55.89 |
> | 256       | 0.0                  | 0.0                        | 84.78     | 82.01            | 31.47 | 56.40 |
> | 256       | 0.1                  | 0.0                        | 84.72     | 81.94            | 31.41 | 56.32 |
> | 256       | 0.0                  | 0.01                      | 84.87     | 82.08            | 31.57 | 56.45 |
> | 256       | 0.0                  | 0.05                      | 84.78     | 81.97            | 31.47 | 56.39 |
> | 256       | 0.0                  | 0.1                        | 84.65     | 81.92            | 31.28 | 56.25 |
> | 256       | 0.0                  | 0.2                        | 84.48     | 81.77            | 30.95 | 56.01 |
>
> > "In Figure 2, do you have results for full fine-tuning?"
>
> Due to storage constraints, we had to delete intermediate checkpoints when using full-finetuning (note that each copy of the model in bf16 takes up to around 13GB). For the final version, we will rerun the best configuration and obtain the necessary results.

---

### Official Review · Reviewer_G2rv · 2023-08-05

**Typos Grammar Style And Presentation Improvements:** Kudos to the authors for a well-writt…
**Soundness:** 3

**Excitement:**

3: Ambivalent: It has merits (e.g., it reports state-of-the-art results, the idea is nice), but there are key weaknesses (e.g., it describes incremental work), and it can significantly benefit from another round of revision. However, I won't object to accepting it if my co-reviewers champion it.

**Paper Topic And Main Contributions:**

This paper studies the impact of finetuning LLMs (LLAMA-7B only) for the translation task, on the translation capabilities of the LLMs, mainly the general translation quality (measured over 10 directions, both E-X and X-E on the Flores-200 test sets) and few-shot domain adapted translation quality (measured on 4 domain-specific test sets: Medical, Law, TICO and WMT-Chat). The key finding is that while fine-tuning (with and without LoRa) improves general translation quality on Flores-200, with LoRa fine-tuning only slightly behind full finetuning performance; the domain adaptation abilities of the model through few-shot learning gets severely impacted. The authors then finetune LLAMA with few-shot examples directly and this recovers the few-shot performance of the finetuned, which is evident in improved domain adaptation performance.

**Questions For The Authors:**

1. Where are the baseline numbers on few-shot when compared with zero-shot on the pre-trained model itself? Knowing these numbers per test sets (both Flores-200 and domain test sets) is crucial for understanding how much drop in few-shot translation quality is observed post-finetuning.
2. Why limit the finetuning data to 160K in Figure 2? Did you stop observing gains after this or was it the maximum data used for finetuning?
3. What was the nature of hallucinations observed in Table 1? Is it output in the wrong language or mainly detached ('Curious Case of hallucinations in NMT', NAACL'21) or overgenerated outputs ('Finding-Memo', EMNLP'22)? Could you quantify it into a few categories based on the hallucination literature (wrong-lang, detached, oscillatory, etc.)?

**Reasons To Accept:**

1. This is a timely paper, with not very comprehensive but interesting and logical results, and can help practitioners help leverage competitive open source LLM implementations such as LLAMA for the task of translation.

**Reasons To Reject:**

1. Non-Comprehensive Experiments: The biggest problem with the paper as it currently stands is that all of the conclusions are made solely based on one 7B LLM, this makes adjudicating the scientific merit of the conclusions very difficult. While the results are quite intuitive, it would have been very interesting to vary the LLMs across two dimensions: scale (at least upto 13B) and diversity (more LLM instances such as MPT, Falcon, Bloom, etc.).
2. Limited Evaluation: The evaluations report only one metric, the reference based COMET-22 metric. LLMs such has PALM, GPT-3.5 have shown to significantly impacted by reference bias and as such reference-free QE metrics such as COMET-KIWI should also be included to understand the impact of finetuning. Similarly, while Flores-200 is a useful test set, I am surprised the authors did not conduct any experiments on WMT-22 test sets. Doing these experiments is crucial since a lot of existing literature has benchmarked results on WMT-22 (e.g. Hendy et-al, How Good are GPTs at MT?) and the absence of any comparisons with the state-of-the-art makes the evaluation look less impressive. Similarly, the baseline numbers on few-shot translation performance on Flores-200 test set is missing in the paper, knowing this number is crucial, since a lot of the mistakes in the zero-shot mode is due to the model not knowing the appropriate output space ('Rethinking the role of demonstrations', Min et al, 2022) and hence producing translations in the wrong language and I am curious if this is the key benefit of finetuning itself.

**Reproducibility:**

4: Could mostly reproduce the results, but there may be some variation because of sample variance or minor variations in their interpretation of the protocol or method.

**Reviewer Confidence:**

5: Positive that my evaluation is correct. I read the paper very carefully and I am very familiar with related work.

---

> ### Author Rebuttal · Authors · 2023-08-28
>
> Thank you for your review.
>
> > "(...) it would have been very interesting to vary the LLMs across two dimensions: scale (at least upto 13B) and diversity (more LLM instances such as MPT, Falcon, Bloom, etc.)."
>
> We recognize that only performing experiments on the 7B LLaMA model may limit the generalizability of our results. We provide a comprehensive answer to your concerns regarding this in the response to R2, who also shared the same concerns. We will make sure to include this analysis in the final version.
>
> > "The evaluations report only one metric, the reference based COMET-22 metric. LLMs such has PALM, GPT-3.5 have shown to significantly impacted by reference bias and as such reference-free QE metrics such as COMET-KIWI should also be included to understand the impact of finetuning."
>
> Please note that we reported results with COMETKiwi in appendix E, as well as BLEU and chrF for completeness. The trends are similar to the ones obtained with COMET-22. We will give more emphasis to the results with the other metrics in the final version.
>
> > "Similarly, while Flores-200 is a useful test set, I am surprised the authors did not conduct any experiments on WMT-22 test sets."
>
> This is a great suggestion. We tested our models on the WMT-22 test set and found that our trends hold across all metrics, as shown below. These results will be included in the camera ready version.
> | Model                  | Context   | COMET | COMETKiwi | BLEU | chrF   |
> | ------------------- | ----------- | --------: | -------------: | ------: | ------: |
> | FT w/o few-shot | Zero-shot | 79.44    | 76.01            | 23.72 | 46.23 |
> |                            | Few-shot  | 79.07    | 75.31            | 23.24 | 45.63 |
> | FT w/ few-shot   | Zero-shot | 80.16    | 76.72            | 24.58 | 47.09 |
> |                            | Few-shot  | 80.23    | 76.77            | 24.63 | 47.27 |
>
> > "Where are the baseline numbers on few-shot when compared with zero-shot on the pre-trained model itself?"
>
> This is a good question. In our paper, both in Figure 1 and Figure 2, we report results with the pretrained model using few-shot, by randomly sampling examples from the development sets (see L135, L143). We will make sure that this is more explicit in the camera ready version.
>
> We did not report zero-shot performance of the pretrained model since it is far behind, particularly in en-xx directions. As such, we considered it more interesting to compare the finetuned models with the pretrained model prompted with few-shot. In any case, the zero-shot results are shown below. We will add these results in the final version (possibly in Appendix).
>
> | Language Pair | COMET | COMETKiwi | BLEU | chrF   |
> | ---------------- | ---------: | -------------: | ------: | ------: |
> | de-en              | 87.23      | 82.99           | 36.77 | 62.34 |
> | en-de              | 71.56      | 57.87           | 13.99 | 38.73 |
> | fr-en                | 87.04      | 84.61           | 37.27 | 62.79 |
> | en-fr                | 76.54      | 65.17           | 23.43 | 47.02 |
> | nl-en               | 84.60      | 83.59           | 25.42  | 53.94 |
> | en-nl               | 73.99      | 61.17           | 12.45  | 38.88 |
> | pt-en              | 87.26      | 83.21            | 41.14 | 65.78 |
> | en-pt              | 74.60      | 55.91            | 17.46 | 40.46 |
> | ru-en              | 83.85      | 82.03            | 28.37 | 57.00 |
> | en-ru              | 57.63      | 46.48            | 7.16   | 17.48 |
> | zh-en             | 76.74      | 77.46            | 12.54 | 34.99 |
> | en-zh             | 49.12      | 45.10            | 4.54   | 7.20  |
>
> > "Why limit the finetuning data to 160K in Figure 2? Did you stop observing gains after this or was it the maximum data used for finetuning?"
>
> Regarding our training procedure, we stopped training because the validation loss plateaued, suggesting that the model had converged, which was supported by the fact that the COMET scores in Figure 2 stabilized.
>
> > "What was the nature of hallucinations observed in Table 1?"
>
> It is true that we did not elaborate on the nature of hallucinations, but we will do so in the CR version. Briefly, we observed that the model generates hallucinations of different categories. In particular, it generates both detached (fully and strongly) and oscillatory hallucinations, and can also generate off-target translations. One common case is that the model copies from the instruction (either the source or from the examples). We will report this information in the camera ready version.

---

### Official Review · Reviewer_mF6X · 2023-08-10

**Soundness:** 3

**Excitement:**

3: Ambivalent: It has merits (e.g., it reports state-of-the-art results, the idea is nice), but there are key weaknesses (e.g., it describes incremental work), and it can significantly benefit from another round of revision. However, I won't object to accepting it if my co-reviewers champion it.

**Paper Topic And Main Contributions:**

This paper provides an analysis into the tradeoffs between few-shot prompting and finetuning for adapting large language models to machine translation.



1/ Adapter-based finetuning matches full finetuning performance with 50 times fewer parameters


2/ Finetuning compromises few-shot performance and the model's ability to adapt to new domains


3/ The authors introduce few-shot examples during finetuning to bring the performance up for few-shot use cases

**Reasons To Accept:**

Experiment results show that FT w/ few-shot is generally beneficial for improving the translation quality when used with few-shot examples

**Reasons To Reject:**

1/ The study is lacking analysis on finetuning with in-domain translation examples vs. general-domain translation examples



The reason is that we might want to analyze whether this fine-tuning process is just specializing the LLAMA model to a translation model





2/Since the main purpose of performing few-shot inference is to adapt the domain of translation. The study is lacking concrete insights on whether the domain-specific translation features are conceretely improved.



For example, when translating with Meidcal/Law, was the model able to absorb the domain knowledge shown in the few-shot examples, such as domain-specific terms, expressions, and styles?

**Reproducibility:**

3: Could reproduce the results with some difficulty. The settings of parameters are underspecified or subjectively determined; the training/evaluation data are not widely available.

**Reviewer Confidence:**

3: Pretty sure, but there's a chance I missed something. Although I have a good feel for this area in general, I did not carefully check the paper's details, e.g., the math, experimental design, or novelty.

---

> ### Author Rebuttal · Authors · 2023-08-28
>
> Thank you for your review.
>
> > "The reason is that we might want to analyze whether this fine-tuning process is just specializing the LLAMA model to a translation model"
>
> Indeed, in this work we aimed to explore how to finetune large language models to machine translation. Our main goal was comparing both in-context learning and finetuning methods, and their overall impact when adapting LLMs for translation.
>
> > "For example, when translating with Meidcal/Law, was the model able to absorb the domain knowledge shown in the few-shot examples, such as domain-specific terms, expressions, and styles?"
>
> This is a great question. We followed previous work that explored domain adaptation through in-context learning (Agrawal et al., 2022) and only reported performance in terms of translation quality. However, verifying whether the model is indeed “adapting” is relevant and we believe our paper can benefit from this analysis. We have analyzed the translations generated in this setting and did find examples in which the model is adapting and using domain-specific terms and styles (see examples below). What is more, perhaps surprisingly, we also found examples in which the introduction of these few-shot examples proved to be harmful (e.g., few-shot examples can introduce hallucinations — see Table 7 in Appendix C for examples of such pathological translations). We will make sure to add this more in-depth analysis in the final version.
>
> Examples of adaptation:
>
> Source: bezeichnet "genetisch veränderte Futtermittel" Futtermittel, die GVO enthalten, daraus bestehen oder hergestellt werden;
>
> Reference: "genetically modified feed" means feed containing, consisting of or produced from GMOs;
>
> Translation with Zero-shot: "Genetically modified feed" means feed containing GVO, derived from GVO or produced from GVO;
>
> Translation with Few-shot: "genetically modified feed" means feed containing, consisting of or produced from GMOs;
>
> Note how in this case the model correctly translates the terminology GVO to GMOs (Genetically Modified Organisms).
>
> --------------------------------------------------------------------------------
>
> Source: VERORDNUNG (EG) Nr. 538/2000 DER KOMMISSION
>
> Reference: COMMISSION REGULATION (EC) No 538/2000
>
> Zero-shot: (EG) No 538/2000 OF THE COMMISSION
>
> Few-shot: COMMISSION REGULATION (EC) No 538/2000
>
> Note how the model is able to correctly order the words in the translation.

---

### Official Review · Reviewer_3izU · 2023-08-11

**Typos Grammar Style And Presentation Improvements:** Page 2, Line 087, there is a misplace…
**Soundness:** 3

**Excitement:**

2: Mediocre: This paper makes marginal contributions (vs non-contemporaneous work), so I would rather not see it in the conference.

**Paper Topic And Main Contributions:**

The paper initially compared the performance of a fully fine-tuned large language model (LLM) and an adapter-based LLM in the context of a machine translation task. The results showed that few-shot learning is not beneficial for the LLM after fine-tuning. Subsequently, the authors introduced few-shot samples into the instructions during the fine-tuning stage. The experiments show that their method can restore the effectiveness of in-context learning.

**Questions For The Authors:**

1. How about using larger foundational models (e.g., 13B and others)?

2. Why do the three length distributions in Figure 4 appear the same?

3. Why does fine-tuning with few-shot examples improve the performance of zero-shot learning? Your explanation is that "the model is extracting helpful information from the examples." However, I find this explanation less convincing, considering the LLaMA's strong capability to extract information.


**Reasons To Accept:**

1. The paper is straightforward and easy to understand.

2. The finding that the fine-tuned LLM cannot benefit from in-context learning is interesting.


**Reasons To Reject:**

1. The contribution is limited, even for a short paper. Regarding the first finding, some open-source projects (e.g., Stanford Alpaca and Alpace-Lora) have demonstrated the effectiveness of finetuning based on the adapter method. The second finding is interesting, but it lacks in-depth analysis and only presents a result. As for the third finding, the in-context learning, which is restored by the proposed method, only shows slight improvement in Figure 3.

2. The analysis is not sufficiently thorough, and more experiments are needed, particularly on larger LLMs.


**Reproducibility:**

3: Could reproduce the results with some difficulty. The settings of parameters are underspecified or subjectively determined; the training/evaluation data are not widely available.

**Reviewer Confidence:**

3: Pretty sure, but there's a chance I missed something. Although I have a good feel for this area in general, I did not carefully check the paper's details, e.g., the math, experimental design, or novelty.

---

> ### Author Rebuttal · Authors · 2023-08-28
>
> Thank you for your review.
>
> > "Regarding the first finding, some open-source projects (e.g., Stanford Alpaca and Alpace-Lora) have demonstrated the effectiveness of finetuning based on the adapter method."
>
> Our paper differs from the Alpaca models as they focus on how to train large language models to follow instructions as a means to generalize to arbitrary tasks. The purpose of our paper is rather to study how large language models can be specialized to machine translation. Note that none of the aforementioned Alpaca projects test efficient finetuning techniques in this specific scenario. One important challenge is that our task is highly multilingual and that LLaMA is highly English-centric. Thus, we find it noteworthy to point out that it is possible to capture the multilinguality aspect of machine translation with efficient finetuning.
>
> > "The second finding is interesting, but it lacks in-depth analysis and only presents a result."
>
> Our second finding reports that finetuning language models degrades their few-shot capabilities. This is an important limitation of current methods, which we believe can be very useful for researchers and practitioners. Note also that our results are consistent across several domains. We welcome any suggestions you may have for deepening the analysis.
>
> > "As for the third finding, the in-context learning, which is restored by the proposed method, only shows slight improvement in Figure 3."
>
> While the gains are relatively small, they are consistent across the several domains. The main take-away of that experiment is that with this method we are able to recover in-context learning abilities, a very important characteristic of language models. This shows that finetuning and in-context learning can coexist.
>
> > "How about using larger foundational models (e.g., 13B and others)?"
>
> Thank you for the suggestion. For the camera ready version, we will run our experiments on the 13B LLaMA model. In the meantime, we have the following preliminary results. We have found that, similarly to LLaMA-7B, the training of the 13B version also seems to plateau around the same number of training examples. As for the results in Table 3, we found that the overall trends reported in the paper seem to hold for the 13B version, with a consistent degradation (as evaluated with COMET and COMETKiwi; we report these results in the tables below) of few-shot performance for finetuned models and a restoration of this performance when finetuning with examples. We will include more detailed results in the final version.
>
> Results with COMET:
> | Model                 | Context   | Flores | WMT | Law | Medical | Tico | Chat |
> | ------------------- | ----------- | ------: | ------: | ----: | --------: | -----: | -----: |
> | FT w/o few-shot | Zero-shot | 85.83 | 80.99 | 84.09 | 82.65   | 83.73 | 87.70 |
> |                            | Few-shot  |85.84  | 80.90 | 83.87 | 82.59   | 83.63 | 87.95 |
> | FT w/ few-shot   | Zero-shot | 86.11 | 81.51 | 84.47 | 82.97   | 83.97 | 87.51 |
> |                            | Few-shot  | 86.12 | 81.51 | 84.50 | 82.99   | 84.04 | 87.67 |
>
> Results with COMETKiwi:
> | Model                 | Context   | Flores | WMT | Law | Medical | Tico | Chat |
> | ------------------- | ----------- | ------: | ------: | ----: | --------: | -----: | -----: |
> | FT w/o few-shot | Zero-shot | 83.06 | 77.54 | 81.40 | 81.14  | 84.36 | 80.86 |
> |                            | Few-shot  | 82.96 | 77.36 | 81.03 | 80.95  | 84.20 | 80.83 |
> | FT w/ few-shot   | Zero-shot | 83.31 | 78.05 | 81.65 | 81.32  | 84.65 | 80.82 |
> |                            | Few-shot  | 83.30 | 78.11 | 81.72 | 81.23  | 84.71 | 80.93 |
>
> > "Why do the three length distributions in Figure 4 appear the same?"
>
> The length distributions measure the number of tokens for the reference translations and the translations generated with both finetuning approaches. The fact that they are similar indicates that the translations generated by the finetuned models are similar in length to the reference translation, suggesting that the models do not overgenerate and learn when to stop. Note that this property is desirable — as it removes any need to post-process/parse the model output — and is not found prior to finetuning (prompting the pretrained model with few-shot exemples leads to overgeneration, as shown in Figure 4). In the camera ready version, we will make this more clear.

---

### Official Review · Reviewer_ceRx · 2023-08-11

**Soundness:** 4

**Excitement:**

4: Strong: This paper deepens the understanding of some phenomenon or lowers the barriers to an existing research direction.

**Missing References:**

Line 087: Europarl: https://aclanthology.org/2005.mtsummit-papers.11/ Paracrawl: https://aclanthology.org/2020.acl-main.417/ please check and add necessary references for all the data sources used.

**Paper Topic And Main Contributions:**

The authors point out the problem that fine-tuning degrades few-shot performance when using LLMs for machine translation. They propose a method to include few-shot examples in the adaptor fine-tuning process, which boost MT performance with fine-tuning while keeping the few-shot capabilities of LLMs. Fine-tuning with few-shot examples improves translation performance, recovers few-shot capabilities, does not suffer from overgeneration, and reduces the rate of hallucination.

**Questions For The Authors:**

This is just out of curiosity - not sure whether using COMETKiwi for quality filtering and using COMET as the main metric would be a problem. Do BLEU scores and other metrics show the same trends?

**Reasons To Accept:**

This work presents an innovative approach to harvest the benefits of fine-tuning and few-shot learning by fine-tuning the LLM with few-shot examples. The problem that the authors are trying to solve - the trade-off between fine-tuning and few-shot learning - is a very important problem with LLMs. The experiments in this work are comprehensive and support the claims made by the authors. I also appreciate the nicely designed figures.

**Reasons To Reject:**

- This paper can benefit from some editing and re-organization.

- The comparison between fine-tuning, adaptor fine-tuning, and LoRA does not seem very relevant to the main point of this work, consider moving this to the appendix when re-organizing the paper.

- It would be nice to see more ways to combine fine-tuning with few-shot examples, why was uniform sampling used? There does not seem to be any motivation or empirical experiments to explain this choice.

**Reproducibility:**

4: Could mostly reproduce the results, but there may be some variation because of sample variance or minor variations in their interpretation of the protocol or method.

**Reviewer Confidence:**

3: Pretty sure, but there's a chance I missed something. Although I have a good feel for this area in general, I did not carefully check the paper's details, e.g., the math, experimental design, or novelty.

**Typos Grammar Style And Presentation Improvements:**

Figure 3: including the model performance with (1) only fine-tuning and (2) only prompt learning as baselines would better contextualize the results.

Line 087: red comma

Line 220: out-of-English

---

> ### Author Rebuttal · Authors · 2023-08-28
>
> Thank you for your review.
>
> > “This paper can benefit from some editing and re-organization” (...) “The comparison between fine-tuning, adaptor fine-tuning, and LoRA does not seem very relevant to the main point of this work”
>
> You are right that the study on the impact of finetuning and few-shot prompting for adapting large language models is the main contribution of this paper. However, we believe our findings about the effectiveness of efficient adaptation are also relevant and can be highly valuable to the research community due to the computational savings they enable. We will, nevertheless, consider these editing improvements in the final version.
>
> > “It would be nice to see more ways to combine fine-tuning with few-shot examples, why was uniform sampling used?”
>
> This is a good question. In fact, we also experimented with considering 50% of the dataset as zero-shot and the remaining 50% containing few-shot instances, uniformly sampling between 1 and 5 examples. This strategy leads to results similar to the ones in the paper, which we will include in the camera ready version. Regarding sampling of examples, our approach is similar to that of Min et al., 2022, which uniformly sampled from the training set. We will make sure to cite this paper in the final version. We recognize that other example selection strategies could bring benefits and deem this an interesting future work direction.
>
> > “[...] Do BLEU scores and other metrics show the same trends?”
>
> This is an important consideration. In our submitted version, we have provided results with several other reference-based and reference-free metrics (CometKiwi, BLEU, and chrF) in Appendix E. Importantly, the trends hold across all metrics. We will make sure to highlight this in the final version.
>
> Finally, we would like to thank you for pointing out the missing references and suggesting presentation improvements. We will make sure to consider them in the camera ready version.

---

### Meta-Review · Area_Chair_wZGV · 2023-09-17

**Recommendation:** 3

**Metareview:**

This paper shows that finetuning with LoRA-based SFT is a good choice for improving the translation performance of LLaMa 7B. It also shows that such a way of finetuning may harms the few-shot translation ability and can be refined by introducing few-shot samples into finetuning. The experiments were conducted on datasets of 10 language pairs.

Fortunately, we have 5 reviewers for this submission. The reviewers agree that this paper investigates interesting issues, and some of the results could be of interest to the community, especially given that tuning LLMs for downstream tasks plays an important role in LLM applications. However, some of the reviewers have concerns about the novelty and experiments. Also, the base model selection is another concern.

My major concern is that the claims here should be examined carefully. For example, the author states that finetuning is harmful to few-shot translation. This might not be a solid conclusion because the author did not finetune the LLM in a few-shot manner. As usual, to achieve good translation abilities for LLMs, we need to use SFT to involve few-shot and zero-shot promotes, rather than zero-shot promotes. Therefore, the finding here is due to the way the author chooses, but not the conclusion we can draw in a general setup. A related problem is that the author uses a relatively large number of samples to tune the model. As is pointed out in related work, more tuning data is not always helpful, and the resulting model tends to overfit this data in many cases.

Another concern is that the use of LLaMa 7B narrows the scope of this work. I basically agree with Reviewer G2rv in that LLaMa is not the best choice for translation. I suspect that the conclusion may change if we use multi-lingual LLMs and/or increase the model size. In this sense, the results are highly dependent on the LLM used in this work.

---

### Decision · Program_Chairs · 2023-10-07

**Decision:**

Accept-Findings

**Comment:**

This paper shows that finetuning with LoRA-based SFT is a good choice for improving the translation performance of LLaMa 7B. It also shows that such a way of finetuning may harms the few-shot translation ability and can be refined by introducing few-shot samples into finetuning. The experiments were conducted on datasets of 10 language pairs.

Fortunately, we have 5 reviewers for this submission. The reviewers agree that this paper investigates interesting issues, and some of the results could be of interest to the community, especially given that tuning LLMs for downstream tasks plays an important role in LLM applications. However, some of the reviewers have concerns about the novelty and experiments. Also, the base model selection is another concern.

My major concern is that the claims here should be examined carefully. For example, the author states that finetuning is harmful to few-shot translation. This might not be a solid conclusion because the author did not finetune the LLM in a few-shot manner. As usual, to achieve good translation abilities for LLMs, we need to use SFT to involve few-shot and zero-shot promotes, rather than zero-shot promotes. Therefore, the finding here is due to the way the author chooses, but not the conclusion we can draw in a general setup. A related problem is that the author uses a relatively large number of samples to tune the model. As is pointed out in related work, more tuning data is not always helpful, and the resulting model tends to overfit this data in many cases.

Another concern is that the use of LLaMa 7B narrows the scope of this work. I basically agree with Reviewer G2rv in that LLaMa is not the best choice for translation. I suspect that the conclusion may change if we use multi-lingual LLMs and/or increase the model size. In this sense, the results are highly dependent on the LLM used in this work.